# Colonizing the clinic: tracking bacterial succession and longitudinal dynamics in five new hospital departments over an entire year

Viktoria Weinberger,[1] Charlotte Neumann,[1] Christina Kumpitsch,[1] Stefanie Duller,[1] Tejus Shinde,[1] Polina Mantaj,[1] Laura Schmidberger,[1] Tamara Zurabishvili,[1] Isolde Halmer,[1] Marina Cecovini,[1] Simone Vrbancic,[1] Kathrin Pepper,[2] Eva Schmon,[2] Julian Wenninger,[3] Lars-Peter Kamolz,[4] Gerald Sendlhofer,[4] Kaisa Koskinen,[1] Christine Moissl-Eichinger,[1,5] Alexander Mahnert[1]

**ABSTRACT** The development of hospital-associated microbial communities over time remains poorly characterized, particularly in terms of how microbial populations dynamically respond to changes in building function, the integration of molecular and cultivation-based data, and the early identification of intervention points for flexible, adaptive microbial control strategies. In this longitudinal study, we investigated microbiome dynamics across five newly built departments at the University Hospital of Graz, Austria, over one year. Surface samples were collected at seven time points: before and after hospital operation started. Alpha and beta diversity analyses revealed a distinct two-phase microbial transition, marked by an initial disruption followed by a gradual homogenization of microbial communities. The strongest driver of community change was the arrival of patients, which led to a significant shift in both diversity and taxonomic composition. While early time points were dominated by environmental taxa such as *Acinetobacter* and *Pseudomonas*, human-associated genera like *Staphylococcus* and *Corynebacterium* became more prevalent over time, particularly on frequently touched surfaces. Department-specific and surface-specific microbial signatures were observed, with outpatient and transplant departments showing more variability than surgical and intensive care units (ICUs). Propidium monoazide treatment indicated that *Pseudomonas* and *Acinetobacter* may persist as viable community members, whereas *Staphylococcus* and *Corynebacterium* likely reflect frequent human deposition. Cultivation data supported these findings, showing episodic contamination primarily linked to human contact. Phenotypic predictions revealed a decline in aerobic, Gram-negative, and potentially pathogenic bacteria over time, although these trends were less pronounced in the ICU. Together, our findings reveal a longitudinal homogenization of hospital microbiomes driven by human activity and highlight key taxa and surfaces that warrant targeted monitoring to improve hygiene protocols and infection control strategies.

**IMPORTANCE** This study provides crucial insights into how hospital environments transform microbially after new departments open, a process poorly understood until now. We reveal a two-phase microbial shift, starting with environmental bacteria like *Acinetobacter* and *Pseudomonas* before the hospital opens, then rapidly transitioning to human-associated microbes such as *Staphylococcus* and *Corynebacterium* once patients and staff arrive. Our findings highlight that human activity is the strongest driver of these changes, especially on frequently touched surfaces. This work is vital for developing targeted and adaptive hygiene concepts, improving infection control, and ultimately making hospital environments safer for patients and staff by focusing on specific surfaces and microbial groups that warrant continuous monitoring.

**Peer Reviewer** Elisabetta Caselli, University of Ferrara, Ferrara, Italy

Address correspondence to Alexander Mahnert, alexander.mahnert@medunigraz.at.

The authors declare no conflict of interest.

See the funding table on p. 24.

KEYWORDS   human microbiome, hospital infections, DNA sequencing, built environments

Hospitals are vital institutions in the healthcare system, providing a wide array of services aimed at promoting health and well-being. As centers for medical treatment and recovery, hospitals accommodate a high turnover of patients, healthcare workers, visitors, and support staff. This constant flux of human activity makes hospitals complex ecosystems characterized by intense and continuous interactions—both among individuals and between individuals and the built environment, including abiotic surfaces, medical equipment, and water and air systems. In this context, the hospital environment represents a dynamic and heterogeneous microbial habitat and harbors a microbiome that is shaped by a multitude of environmental factors (1–10). These microbial communities contain both commensal microorganisms and opportunistic pathogens of clinical importance, capable of causing healthcare-associated infections (HAIs) (1). It is estimated that in Europe alone, 3.5 million cases of HAIs occur each year, leading to more than 90,000 deaths (11), with poor cleaning of hospital surfaces identified as a major source of HAIs (12).

In general, the importance of the indoor microbiome for human health is increasingly recognized, extending beyond clinical settings to include homes, schools, workplaces, and public buildings (13). This growing interest is driven by the fact that individuals, particularly in urbanized and high-income societies, spend an estimated 90% of their daily lives indoors, making the built environment the principal interface for microbial exposure (8, 14, 15). Evidence suggests that the indoor microbiome can support key health-related processes, such as the microbial colonization and immune development of the infant gut (16–18), or contribute to microbiome recovery after infections or antibiotic treatment (19).

The indoor environment is shaped by numerous factors. This includes human presence and activity (contributing substantially to microbial load through skin, respiratory emissions, clothing, and direct contact), cleaning and disinfection practices, ventilation systems including air filtration, and material composition of surfaces (20). Another important determinant is the level of introduction of environmental microbiomes to the indoor area. In highly controlled or confined environments, such as the International Space Station (ISS), cleanrooms, or certain areas of the hospital, including surgical theaters, transplant units, and intensive care units (ICUs), exogenous microbial influx is minimized by design (21, 22). These settings are maintained under strict microbial control to reduce the risk of contamination or infection, resulting in unique microbial ecosystems that are often characterized by low diversity, human-associated dominance, and selective pressure from disinfection and antibiotic exposure (21, 22).

Also, humans play a substantial role in the microbiome composition of a hospital microbiome (5, 8, 14, 23–26). They are often the primary source of microorganisms found in these environments, with microbial exchange occurring bidirectionally between humans and the environment (27–32). Frequently touched surfaces reflect the microbial profiles of the individuals who interact with them (1, 33), and fluctuations in human occupancy directly influence microbial composition (1, 8, 23, 34). In newly opened hospitals, surface microbiomes initially resemble those of the outdoor environment, but with the presence of patients and staff, these communities shift toward human-associated taxa, particularly from skin and respiratory tract, including signatures of *Corynebacterium*, *Staphylococcus*, *Streptococcus*, and *Acinetobacter* (1, 8, 23, 34). Hospital staff can act as vectors, disseminating microbes throughout the facility, while patients imprint their microbial patterns on their immediate surroundings, leading to increasing similarity between room and occupant microbiomes over time (34).

Longitudinal studies have highlighted that the hospital microbiome is not static but undergoes dynamic changes over time. In some cases, bacterial communities on hospital surfaces remain relatively stable, with dominant taxa persisting despite routine cleaning and disinfection (35). However, upon the opening of a new hospital, the microbial load

on surfaces increases, and the microbial composition shifts from primarily environmental taxa to those associated with humans, particularly skin-related genera. Surfaces within patient rooms, such as bedrails or remote controls, increasingly reflect the microbiota of the occupying patient, with microbial overlap between patients and their immediate environments becoming more pronounced over time (34). Patients initially acquire microbes left by previous occupants or introduced via environmental sources, but as their stay progresses, they imprint their own microbial signature on the surroundings (34). Conversely, when a hospital unit is decommissioned, the relative abundance of human-associated microorganisms declines, while environmental bacteria become more prominent (1, 23, 36). These observations highlight the bidirectional nature of host-environment microbial exchange and the rapid temporal dynamics shaping the hospital microbiome. Within a few weeks of occupancy, departments often develop stable yet distinct microbial communities, with localized colonization patterns and, in some cases, the emergence and spread of antibiotic resistance genes (8, 21, 33, 35, 37).

Given their impact on patient safety, the hospital microbiome is of particular concern due to its role as a potential reservoir for opportunistic pathogens. Even in facilities with rigorous infection prevention protocols, environmental contamination can persist and contribute to the spread of HAIs (4, 5, 21, 38–45). Surfaces touched frequently by staff, patients, and visitors can serve as nodes in transmission pathways, enabling pathogens to spread between individuals or persist in the built environment (46). Besides multidrug-resistant (MDR) bacteria such as methicillin-resistant *Staphylococcus aureus* (MRSA), vancomycin-resistant enterococci (VRE), and carbapenem-resistant Gram-negative bacteria have been detected on hospital surfaces (47–53). Many of these pathogens show alarming levels of antibiotic resistance, which complicates treatment strategies and contributes to poor patient outcomes (21, 38).

Characterizing the microbiome of the hospital environment is therefore crucial for understanding its impact on patient care and infection control. Detailed profiling of these microbial communities not only supports outbreak investigations and improves our knowledge of pathogen reservoirs, but also informs strategies to prevent HAIs and mitigate the spread of resistance (4, 8, 34, 37, 40, 41, 44, 54–56). Importantly, investigating microbial dynamics, colonization patterns, the resistome, and responses to environmental factors such as cleaning, occupancy, and surface materials provides an opportunity to manage these ecosystems more effectively (21).

Despite growing awareness of the role built environments play in shaping microbial communities, the temporal dynamics of microbial colonization in newly opened hospital departments remain poorly understood. In particular, there is limited insight into how microbial populations respond to shifts in building function, such as the transition from construction to clinical use. Existing studies often rely solely on molecular techniques, overlooking the complementary value of using propidium monoazide (PMA) to identify the fraction of intact cells in molecular data or include standard cultivation-based approaches that can still reveal viable and clinically relevant taxa. Moreover, current hygiene strategies tend to be static and generalized, lacking the flexibility to adapt to microbial changes driven by human activity, especially on high-touch surfaces. There is a pressing need to identify early microbial transition points that could serve as targets for proactive and adaptive infection control, rather than reactive interventions. Addressing these gaps is essential for developing precision hygiene concepts that are both ecologically informed and operationally feasible.

In this longitudinal study, we investigat the development of the microbiome in a newly constructed hospital building over the course of one year. We focus on five departments with different functions and patient vulnerability: Ambulatory Care Unit (Amb, an outpatient ward—polyclinic), General and Visceral Surgery (Gen_Surg), Thorax Surgery (Thx), Transplant Surgery (Trans), and the ICU. There, multiple surface types across various time points were examined. Our goal was to characterize how microbial diversity and community structure evolve during hospital operation, identify potential sources of pathogens, and assess how environmental and functional differences

between departments influence microbial composition. Additionally, by integrating viability assessments through PMA treatment, we distinguish between DNA signals from intact versus non-intact cells, allowing deeper insights into the active microbial populations present on surfaces over time. Our findings aim to support future strategies for microbial-aware hospital design, cleaning protocols, and infection prevention.

## MATERIALS AND METHODS

### Study design

The study was performed at the University Hospital of Graz, Austria, in a newly built surgical department including the following five departments: Ambulatory Care Unit (Amb, an outpatient ward- polyclinics), General and visceral Surgery (Gen_Surg), Intensive Care Unit (ICU), Thorax (Thx) and transplant (Trans) surgery. The five departments differ in their confinement levels in means of accessibility for patients and visitors. The ICU department was the most restricted area, whereas Gen_Surg, Thx, and Trans were less restricted, and Amb had the easiest accessibility.

Surface samples were taken at seven different time points. The first sampling (time point 0) took place prior to the opening of the building (unoccupied), and then in increasing intervals in a period of one year (Fig. 1). Sampling sites included patient rooms and bathrooms, occupied with a different number of patients (single, double room, three to four bed room, isolation room, examination room, and recovery room) and non-patient-related rooms like a changing room for the female personnel were sampled. Swab samples were taken from 17 different surfaces, such as door handles, medical workstations, patient beds, the sink, and toilet flush buttons, among others (Fig. 1; Table S1). Wipes (Sterile Wipe LP, Texwipe, Kernersville, NC, USA) were used to sample the floor in the specified sampling locations. Further, propidium monoazide (PMA) treatment was performed to differentiate between intact/viable cells and non-intact/dead cells for eight sample types, namely sink, toilet, bed frame, bed remote control, OP lamp, touch screen, and lightbar above bed.

A total of 1,554 samples (including 1,337 surface samples, 217 controls, including field controls, and PCR controls) were processed throughout the course of this study (Fig. 1; Fig. S1).

### Sampling and sample processing

#### Wipes

The samples were taken as described earlier by Duller et al. (21). The protocol of the European Cooperation for Space Standardization (ECSS-Q-ST-70-55C) was followed for the wipe samples (57). Briefly, floors were sampled using DNA-free pre-moistened wipes (Sterile Wipe LP, Texwipe, Kernersville, NC, USA). Wipes were baked for 24 h at 170°C, moistened with 15 mL sterile DNA-free water (LiChrosolv grade water, Merck KGaA, Darmstadt, Germany), and finally autoclaved (20 min, 121°C) in sterile 50 mL reaction tubes (Sarstedt AG & Co, Nuembrecht, Germany), to ensure full degradation of DNA. In the different sampling locations, approximately 1 m$^2$ of the floor was sampled by moving the wipe horizontally, vertically, and diagonally over the respective floor surface. Field controls were taken by moving a sterile wipe through the air before placing it back into the tube. Wipes were stored at −80°C in sterile 50 mL reaction tubes (Sarstedt AG & Co, Nuembrecht, Germany) until further use. For the extraction of the floor samples' biomass, wipes were thawed at 4°C overnight and then transferred to 250 mL wide-neck flasks (baked at 250°C, 24 h) containing 80 mL Milli-Q grade water (Merck KGaA, Darmstadt, Germany) using baked (250°C, 24 h) tweezers. Next, the wide-neck flasks were manually shaken for one minute and sonicated for two minutes at 240 W and a frequency of 40 kHz (Ultrasons, J.P. selecta, Barcelona, Spain). Floor samples were then concentrated down to approximately 200–250 µL using Amicon Filters (Amicon Ultra-15 [50K NMGG], Merck Millipore Ltd., Tullagreen, IRL, UV sterilized for 1 h) and repeated centrifugation

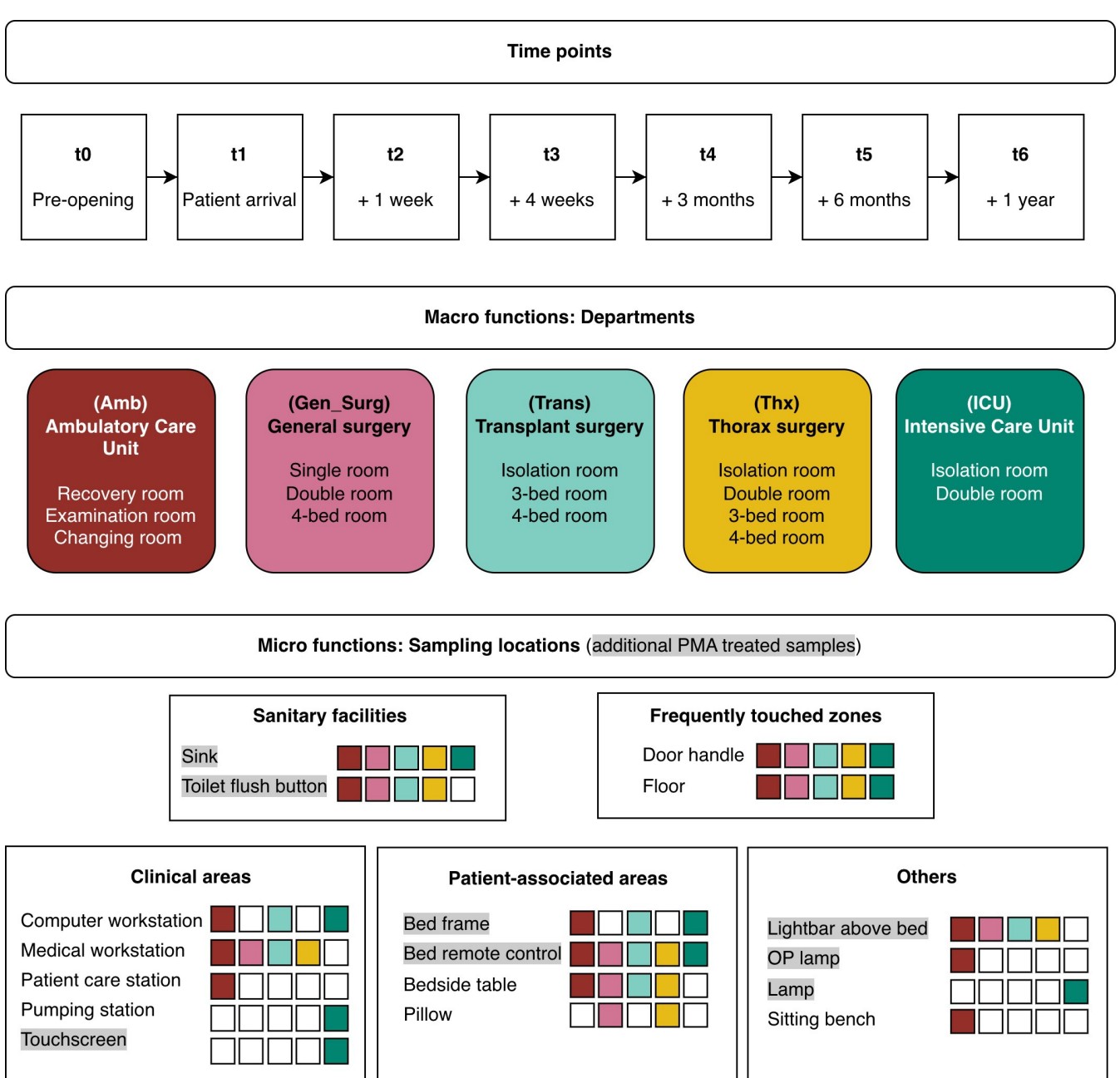

**FIG 1** Overview of study design. Boxes indicate sample locations in each department (white = no samples, gray highlight = additional PMA-treated samples). Abbreviations t0–t6 correspond to time points 0–6.

(3,500 × g, 5 min, 4°C). Amicon filters were rinsed with the concentrated sample to increase the yield output, and then, the sample was transferred to sterile 1.5 mL Eppendorf tubes and stored at −80°C.

*Swabs*

All other surfaces were sampled with swabs (BD BBL Culture Swabs EZ, Copan, Italy), pre-moistened in 0.9% DNA-free NaCl (wt/vol), by rubbing the surface of interest (approximately 10 m², horizontal, vertical, and diagonal with consistent pressure). A subset of surfaces was sampled with two swabs (Fig. 1; Table S1), one sample was used for PMA treatment (described in the next section), the other one for the direct (non_PMA treated) DNA extraction. Until further processing, all samples were immediately stored

at −80°C. To control the sterility of the material used, controls were taken at every time point by moistening the swabs in DNA-free NaCl. All samples were placed on ice directly.

### PMA treatment

Differentiation between dead/non-intact and viable/intact cells was enabled by PMA treatment for a subset of swab surface samples and respective field controls directly after the sampling procedure (Fig. 1; Table S1). The dye PMA intercalates into (free) DNA as long as it is not protected by the membrane of an intact cell. As a result, PCR products will only be generated from the membrane-protected fraction of intact cells in a sample (58). The treatment was carried out as previously described by Duller et al. (21). At first, 250 µL DNA-free 0.9% NaCl and 50 µM PMA (PMA dye, Biotium Inc., Fremont, USA) were added to the sample. The samples were then vortexed and incubated in the dark (RT) for 5 min. Activation of PMA intercalation was initiated by exposing the samples to light using a PMALite LED photolysis device (465–475 nm) (PMA dye, Biotium Inc.) for 15 min. Subsequently, samples were stored at −80°C until DNA extraction.

### Contact plates and cultivation

In addition to the surface sampling for 16S rRNA gene sequencing, Caso RT/RTplus contact plates (casein–soy peptone agar; Merck Millipore, Germany) were used for sampling the surfaces at every time point. Plates were incubated at 36°C for 2 days, before colony-forming units (CFUs) were counted and strains were identified using a Vitek mass spectrometer (Table S7). These samples were processed at the Institute for Hospital Hygiene and Microbiology, Graz, Austria.

## DNA extraction, PCR, and sequencing

DNeasy PowerSoil Kit (QIAGEN GmbH, Hilden, Germany) was used for the DNA extraction of all samples (wipes/floor, swabs, PMA-treated swabs, and field control samples). The manufacturer's instructions were followed, except for the following changes: rather than 250 mg soil, either swab samples (DNA extraction only), swabs and 250 µL DNA-free 0.9% NaCl (PMA treatment), or 250 µL of floor samples (wipes) were added to the Power-Bead Tubes. Furthermore, a MagNA Lyser device (Roche Diagnostics, GmbH, Mannheim, Germany) was used for the bead beating step (2 × 30 s, 6,400 Hz) with a cooling step on ice in between the two sets. For step 5, samples were centrifuged for 2 min at 13,000 × $g$, and 50 µL of solution C6 was added in the final step. For each DNA extraction, a kit control (no template control of potential contaminants in the kit's reagents) was included. Samples were stored at −80°C and further processed.

Amplification of the variable region V4 of the 16S rRNA gene was performed via polymerase chain reaction (PCR) with the universal primers, namely 515F (5′- GTGY-CAGCMGCCGCGGTA- 3′) and 806R (5′- GGGACTACNVGGGTWTCTATT-3′), Ex Taq DNA polymerase (Takara Bio Inc., Japan), and 2 µL of template DNA. Cycling conditions were used according to Caporaso et al. (59, 60), initial denaturation for 3 min at 94°C, 35 denaturation cycles for 45 s at 94°C, annealing 60 s 50°C, extension 90 s 72°C, and final extension for 10 min 72°C.

Library construction and next-generation sequencing (Illumina MiSeq) were performed at the Core Facility Molecular Biology at the Center of Medical Research (Graz, Austria). Sample preparation included the use of a SequalPrep Normalization Plate (Life Technologies, Carlsbad, USA) for the normalization of the PCR products, followed by a PCR to index the samples with unique barcode sequences (61). Furthermore, purification of samples was performed with QIAquick Gel Extraction Kit (QIAGEN GmbH), and validation and quantification were done with Promega Quantus device and Agilent 2100 Bioanalyzer (Agilent, Santa Clara, USA) (61).

## Data processing and analysis

The raw sequencing data were processed with QIIME2 (Version 2022.8) following the provided guidelines in the QIIME2 documentation (https://docs.qiime2.org/) and as previously described by Caporaso et al. (62). DADA2 was used for removal of primers, chimeric sequences, and quality filtering with truncation (-p-trunc-len-f 200 -p-trunc-len-r 150, to ensure high-quality scores greater than $Q$-score 30) and denoising for generating amplicon sequence variants (ASVs) (63). Taxonomic classification was conducted using a classifier trained on the 16S rRNA gene reference sequences from the SILVA database (version 138) (64). To identify potential contaminating ASVs, the R package decontam (v1.22) was used for identification and removal according to the processed controls (https://github.com/benjjneb/decontam) (65). Consequently, 867 out of 49,312 features were excluded from the data set running *iscontaminant* in *prevalence* mode at a threshold of 0.5. As a next step, negative and field controls, and sequences identified as mitochondrial or chloroplast were removed. For normalization, scaling with ranked subsampling (SRS) was run in QIIME2 with $c_{min} = 1,000$. Thereby, 270 samples were removed from the data set due to low counts (Fig. S1). All further analyses were conducted on the cleaned and SRS normalized data set.

## BugBase

BugBase was used to infer phenotypes from representative ASVs (66). To ensure full compatibility with BugBase's default database, representative sequences were also picked using Greengenes (gg_13_8) as a closed reference at 99% similarity within QIIME 1.9.1. Python's pandas and numpy functions were used to merge the Greengenes IDs with the feature table generated with QIIME2 2022.8 (see details above). The run.bugbase.r script was then executed in R-4.4.1 to normalize 16S rRNA gene copy numbers, predicting phenotypes, and plotting thresholds, predictions, and ASV contributions on different Departments, and time as a continuous variable. Up to 1,480 ASVs could be matched to the default BugBase database.

## Statistics and data visualization

For creating the flow chart and study design, the online tool draw.io was used (67). Alpha and beta diversity, line plots, and bubble plots were visualized using the R libraries (ggplot2, tidyverse, dplyr, reshape2, microbiome, ggpubr, and phyloseq). Additionally, the 3D PCoA (principal coordinates analysis) and deicode PCA (principal component analysis) biplots were created with the QIIME2 plugin emperor. Deicode performs robust Aitchison PCA on compositional data. The top 10 taxa with the highest feature loadings were displayed as vectors in biplots, indicating their role as potential drivers of microbial community composition across samples. Relative abundance of sequencing data were generated with the R package Microbiome explorer (68). Differential abundance between macro-functional and micro-functional levels, and the cultivation data were tested with the R package MaAsLin2, including time and departments or sampling locations as fixed effects and different room types as random effects (69). For supervised machine-learning models, the QIIME2 plugin q2-sample-classifier was used to highlight taxa that contributed most significantly to model predictions on macro-functional and micro-functional levels (70).

## RESULTS AND DISCUSSION

### Microbial alpha diversity followed a longitudinal homogenization pattern

Over the period of one year, the development of the hospital microbiome across five newly built departments at the University Hospital of Graz, Austria was investigated. Various surface samples were collected covering seven time points: the baseline (t0), before the departments became operational, and six follow-up points after patient and staff occupation: immediately after opening (t1), 1 week (t2), 4 weeks (t3), 3 months (t4), 6 months (t5), and 1 year later (t6) (Fig. 1).

The covered departments in this study were: Ambulatory Care Unit (Amb, an outpatient ward- polyclinics), Intensive Care Unit (ICU), General and visceral Surgery (Gen_Surg), Thorax Surgery (Thx), and Transplant Surgery (Trans). The departments differed in patient turnover and room access: the ICU had the highest level of access restriction and the longest patient stays. Trans was equipped with an airlock system, whereas Amb had minimal restrictions and only short-term patient visits. In contrast, Gen_Surg and Thx shared a comparable room structure and organization, which facilitates more direct comparisons of their microbial communities.

Seventeen distinct sampling locations across departments were selected to reflect a range of clinical rooms (e.g., isolation, double, and multi-bedrooms, recovery and examination rooms) and functional surfaces. These locations were chosen because they represent sites with different levels of patient contact, hygiene procedures, and environmental exposure, thereby capturing a broad spectrum of microbial inputs within the hospital setting. The sampling included patient-associated areas (e.g., bed frames, remotes, and bedside tables), sanitary facilities (e.g., sinks and toilet flush buttons), and frequently touched zones (e.g., door handles and floors; for a summary of all sampling locations, see Fig. 1).

After DNA extraction, sequencing, raw data quality control, and normalization, a substantial number of samples (270 out of 1,336) did not pass the established quality thresholds and were consequently excluded from downstream analyses. This was expected, as the samples originate from low biomass environments that are frequently cleaned with rigorous detergents that interfere with DNA extraction performance. See Fig. S1 and Table S1 for an overview of sample dropouts.

In this study, we focus on two main functional properties of hospital building operation: the macro-functional level, represented by entire departments, and the micro-functional level represented by individual, smaller-scale (max. 1 $cm^2$ to 1 $m^2$) sampling locations inside the hospital.

At the macro-functional level, microbial diversity between the five departments showed similar longitudinal patterns, and no significant differences could be depicted (Fig. 2A; Shannon diversity, MaAsLin2, departments cross-sectional, $q = 0.83$; Table S2). However, when individual departments were compared over time, distinct differences could be identified: after an initial phase of fluctuation, the microbiome's Shannon diversity stabilized with a mean of 3.29 and a high confidence interval ranging from 1.21 to 5.37 on macro-functional level (Fig. 2B). Therefore, longitudinal changes of the hospital microbiome were split into two phases: before and after the hospital became operational by the patients' arrival. From all departments, Amb and Trans showed the strongest difference in the microbial diversity between t0 and t1 (Shannon diversity, MaAsLin2, Amb: coeff = −0.6, $q \le 0.001$; Trans: coeff = −0.3, $q = 0.01$; Table S2). Briefly, microbial diversity in Amb significantly decreased until t3 (4 weeks after the department got operational) compared to the baseline t0 (Shannon diversity, MaAsLin2, t1: coeff: −0.6, $q \le 0.001$; t2: coeff: −0.6, $q \le 0.001$; and t3: coeff: −0.3, $q = 0.05$; Table S2). Also, the richness first decreased until t3 and then increased at t4 (Amb: richness, MaAsLin2, t1: coeff: −1.8, $q \le 0.001$; t2: coeff: −1.5, $q \le 0.001$; t3: coeff: −0.8, $q = 0.08$; and t4: coeff: 0.7, $q = 0.19$; Table S2). Evenness significantly decreased at t1 and t2 in contrast to t0 (Amb: Pilou's evenness, MaAsLin2, t1: coeff: −0.2, $q \le 0.001$; and t2: coeff: −0.3, $q \le 0.01$; Table S2).

For the ICU, diversity significantly decreases between the period of t2 and t5 in comparison with the baseline t0 (Shannon diversity, MaAsLin2, t1: coeff: −0.6, $q \le 0.001$; t2: coeff: −0.6, $q \le 0.001$; and t3: coeff: −0.3, $q = 0.05$; Table S2).

Still, some differences could be observed between the departments. Shannon diversity only increased for Gen_Surg and ICU after t4, whereas it decreased for the other departments during this period (Fig. 2B). For Amb and Trans, a significant decrease in alpha diversity could be detected from t0 to t1 (Shannon diversity, MaAsLin2, Amb: $P = 1.9e−06$, Trans: $P = 0. 00081$; Table S2), followed by a significant increase to t4 (Amb: $P = $

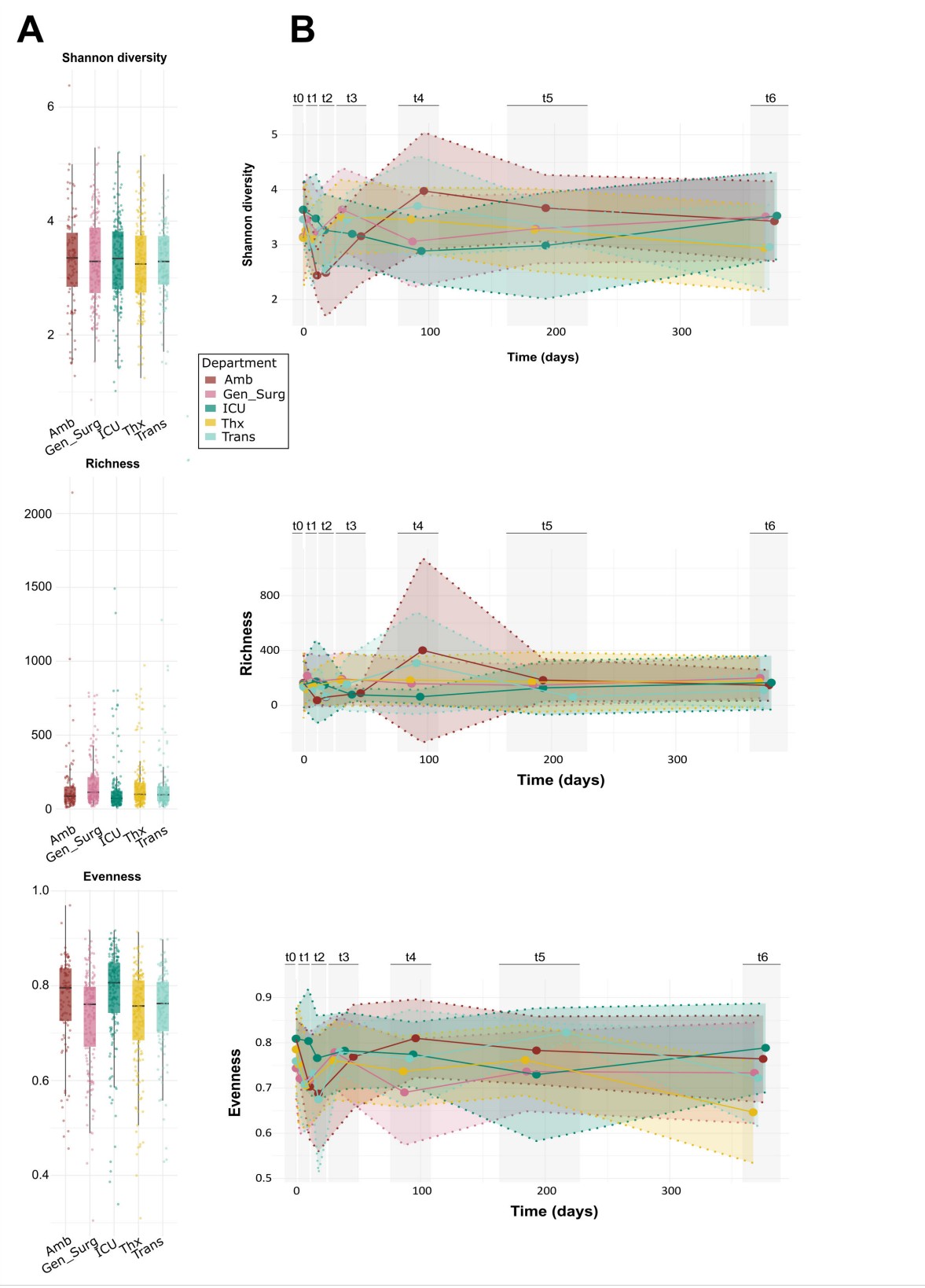

**FIG 2** Alpha diversity at the macro-functional level. (A) Diversity per department: Shannon diversity, richness (observed ASVs), and evenness. (B) Diversity over time for each department.

0.00029, Trans: $P = 0.0012$; Table S2). The same could be observed for the two other alpha diversity metrics, richness and evenness (Table S2).

It seems as if the alpha diversity of the indoor microbiome undergoes some fluctuations at the beginning, followed by a stabilization over time. Similarly, other studies investigating newly opened hospital departments reported significant increases in alpha diversity immediately after patient occupancy, which then stabilized within a few weeks (8, 71).

## The arrival of patients permanently changes the beta diversity of a hospital

Like the alpha diversity patterns above, the beta diversity also followed this two-phase pattern: before and after the hospital became operational. The admission of patients appears to be the strongest driver, as all post-opening time points (t1–t6) differ significantly from the baseline (t0) (Fig. 3A; Bray-Curtis, MaAsLin2, t1: coeff: 1.1, $q = 0.001$; t2: coeff = 1.3, $q = 0.002$; t3: coeff = 0.9, $q = 0.0045$; t4: coeff:1.3, $q = 0.002$; t5: coeff = 1.0, $q = 0.0043$; and t6: coeff = 1.5, $q \leq 0.001$; Table S3).

On a macro-functional level (department level), ICU samples from t1 to t4 and t6 showed significant differences from t0, but there were no significant differences from t1. Thx exhibited a similar pattern, although only t2 and t6 differed significantly from t0. This suggests that microbial dynamics were most strongly impacted immediately after the hospital opened and then stabilized longitudinally (Fig. 3A, Bray-Curtis, MaAsLin2, ICU: t1–t4 and t6: $q \leq 0.05$; Thx: t2 and t6: $q \leq 0.05$; for further details, see Table S3). In contrast, Amb and Trans showed greater fluctuations over time (Fig. 3A).

Hence, the strongest impacts were visible at the beginning (t0 vs t1), before the microbial community stabilized from t2 onwards, indicating a process of longitudinal homogenization.

While alpha diversity did not differ significantly on the macro-functional level (Fig. 2A), clear differences in beta diversity were observed (Fig. 3B). Amb and Trans samples were more similar to each other with a higher variation, while the other three departments (Gen_Surg, Thx, and ICU) formed tighter clusters (Fig. 3B). The greater variability in Amb may be attributed to both temporal fluctuations and poorer sequencing quality. Furthermore, Amb and Trans had a high sample dropout during SRS normalization (Amb: 101 samples, Trans: 57 samples, Fig. S1 and Table S1). These differences may also reflect the distinct functions and activities in each department (e.g., patient contact, usage of different clinical areas, and therefore more or less microbial input [24]), and variation in sampling sites (micro-functional level). Moreover, the higher beta diversity in Amb and Trans could be driven by increased microbial turnover due to shorter patient stays, greater staff movement, or more diverse environmental inputs. In contrast, Gen_Surg, Thx, and ICU may harbor more stable microbial communities due to more uniform patient populations, structured room usage, and consistent cleaning regimens (24). These patterns underscore the role of human activity as a key driver of hospital surface microbiomes and highlight the need to consider both department-specific practices and micro-functional surface characteristics when designing infection prevention strategies (8, 14, 72).

One aspect that can lead to differences between the departments was also the sampling of different rooms and locations (micro-functional level). In total, 16 rooms and 17 distinct sampling locations were analyzed (Fig. 1). Some locations were exclusive to specific departments (such as OP lamp, patient care station, and sitting bench in Amb, or pumping station, touchscreen, and lamp from ICU), while others (e.g., floor, door handle, bed remote control, and sink) were sampled in all departments. Sampling locations were selected to capture a range of environments that differ in contact frequency, proximity to patients, and functional role within the hospital. For example, patient-associated areas (e.g., bed remote control, pillow, bedside table, and bed frame) reflect direct patient contact, frequently touched areas (e.g., door handle and floor) provide insight into transmission pathways, and sanitary facilities (e.g., sink and toilet flush button) represent potential microbial reservoirs influenced by moisture and hygiene-related activities.

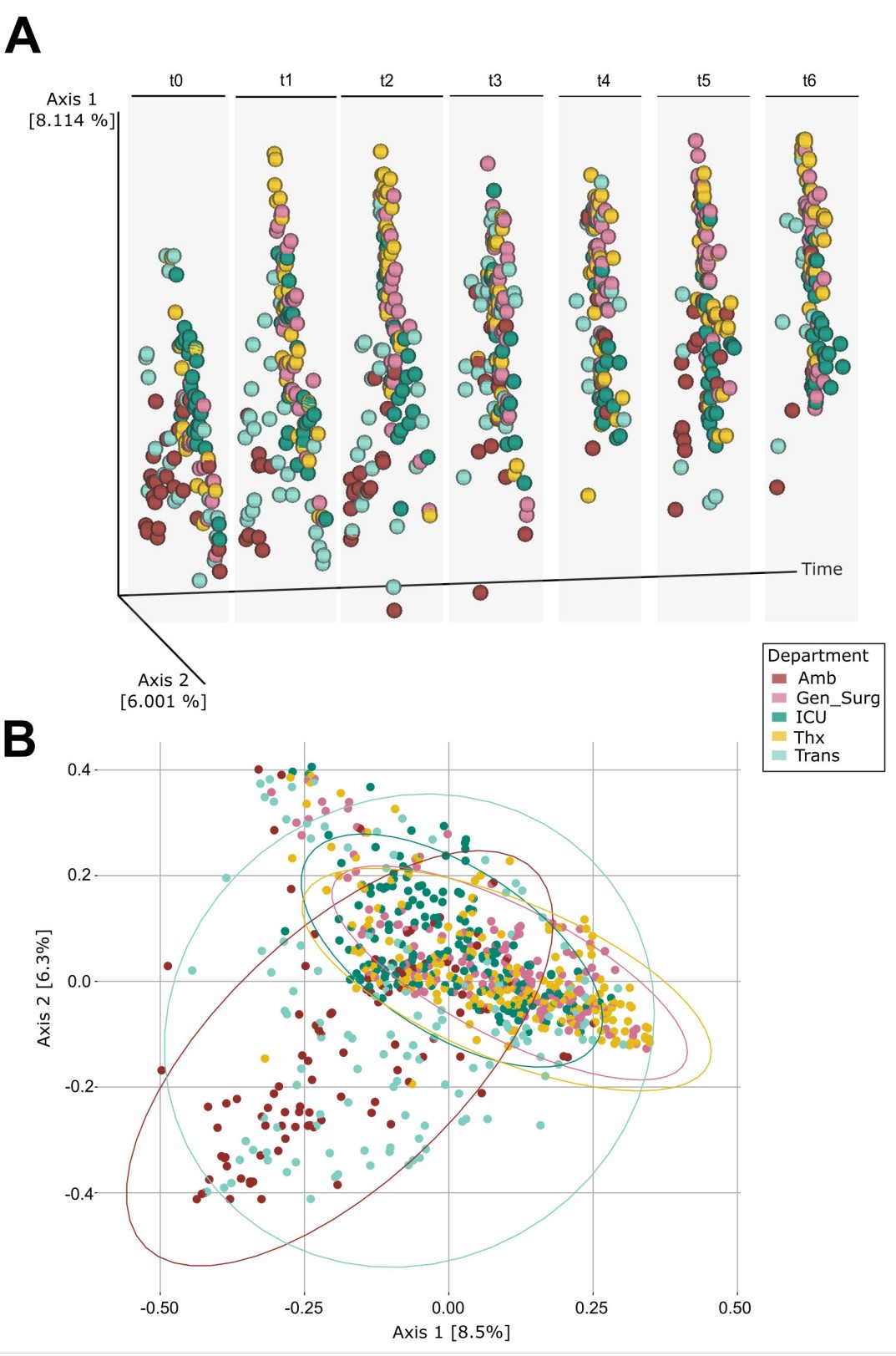

**FIG 3** Beta diversity at the macro-functional level. (A) Department-wise beta diversity shown separately for each time point. (B) Bray-Curtis PCoA of all time points merged per department at ASV level.

For detailed comparative analyses on a micro-functional level, we focused on the four sampling locations shared across all five departments: floor, bed remote control, door handle, and sink (see Fig. S2 and S3 for a complete summary of Shannon diversity across all locations). Floor samples consistently showed the highest Shannon diversity and richness (Fig. 4A; Shannon diversity, MaAsLin2, bed remote control, door handle, and sink: $q < 0.001$; Table S2). Bed remote control and door handle, representing high-contact surfaces and sink samples as a wet environment, displayed similar Shannon diversity and evenness indices (Fig. 4A).

Longitudinal trends of Shannon diversity are shown in Fig. 4B for each of the four common locations. All sampling sites exhibited strong diversity shifts at the beginning, followed by stabilization over time (Shannon diversity, MaAsLin2, Table S2). Floor and door handle samples showed a general increase in Shannon diversity over time. Similar patterns were observed in Amb and Trans departments across all locations. Interestingly, bed remote control samples followed nearly identical diversity trends across all departments. Notably, floor samples had the highest Shannon diversity before hospital opening. In Amb, microbial diversity initially dropped post-opening, then gradually increased again across all four sampling locations (Fig. 4B).

In terms of beta diversity, Amb and Trans continued to cluster slightly apart from the other departments (Fig. 3). ICU, Gen_Surg, and Thx showed more consistency. Overall, the sampling site appeared to have a greater influence on community composition than the department itself ($R^2$ averages of pairwise Adonis tests from all shared sampling sites: 0.05, FDR $P$adjust 0.004; vs $R^2$ of pairwise Adonis tests from all departments: 0.02, FDR $P$adjust 0.002).

The five hospital departments exhibited distinct beta diversity patterns across different sampling locations, with certain surfaces (such as floor, door handle, bed remote control, and sink) showing stronger clustering by their source department than others (Fig. 4C; Table S3). Again, we focused on the four locations sampled in all five departments (floor, door handle, bed remote control, and sink). Floor samples from Amb formed a largely separate cluster compared to other departments, except for some overlap at t0 and t5, suggesting temporal shifts in microbial diversity (Fig. 4C). This aligns with trends observed for Shannon diversity, where microbial composition at t0 and t5 appeared more similar across departments.

For bed remote controls, samples from ICU, Thx, and Gen_Surg exhibited highly similar beta diversity distances, forming a distinct cluster in PCoA ordinations (Fig. 4C). In contrast, Trans and Amb again showed more distinct profiles, with samples from t0 and t2 positioned closer together, suggesting temporal stability or similar early colonization patterns. Door handle samples followed a similar trend, where Gen_Surg, ICU, and Thx clustered closely, while Amb and Trans remained more distinct, though still exhibiting some overlap with the other departments (Fig. 4C). Sink samples displayed the highest similarity in beta diversity across all departments, indicating that sinks, regardless of location, may serve as a shared microbial reservoir, possibly influenced by environmental moisture and water-associated taxa (Fig. 4C [73–76]).

Overall, across the four sampling locations, Gen_Surg, ICU, and Thx departments displayed greater similarity in microbial community composition, whereas Amb and Trans showed more distinct patterns. This suggests that certain departments harbor more comparable microbial communities, potentially due to similar patient demographics, room usage, or environmental factors, while others, such as Amb and Trans, may be shaped by different interactions or cleaning practices (21, 24, 77–79).

## Bacterial taxa follow the purpose of a department in terms of patient care

Only 10 bacterial genera expressed up to ~70% of the entire relative abundance within the data set (Fig. 5A and B). *Corynebacterium*, *Staphylococcus*, *Acinetobacter*, *Pseudomonas*, and *Streptococcus* were the most abundant genera in all departments at all time points. This observation is in line with other key publications in this field (33). *Acinetobacter* and *Pseudomonas*, two environmentally persistent opportunists, dominated

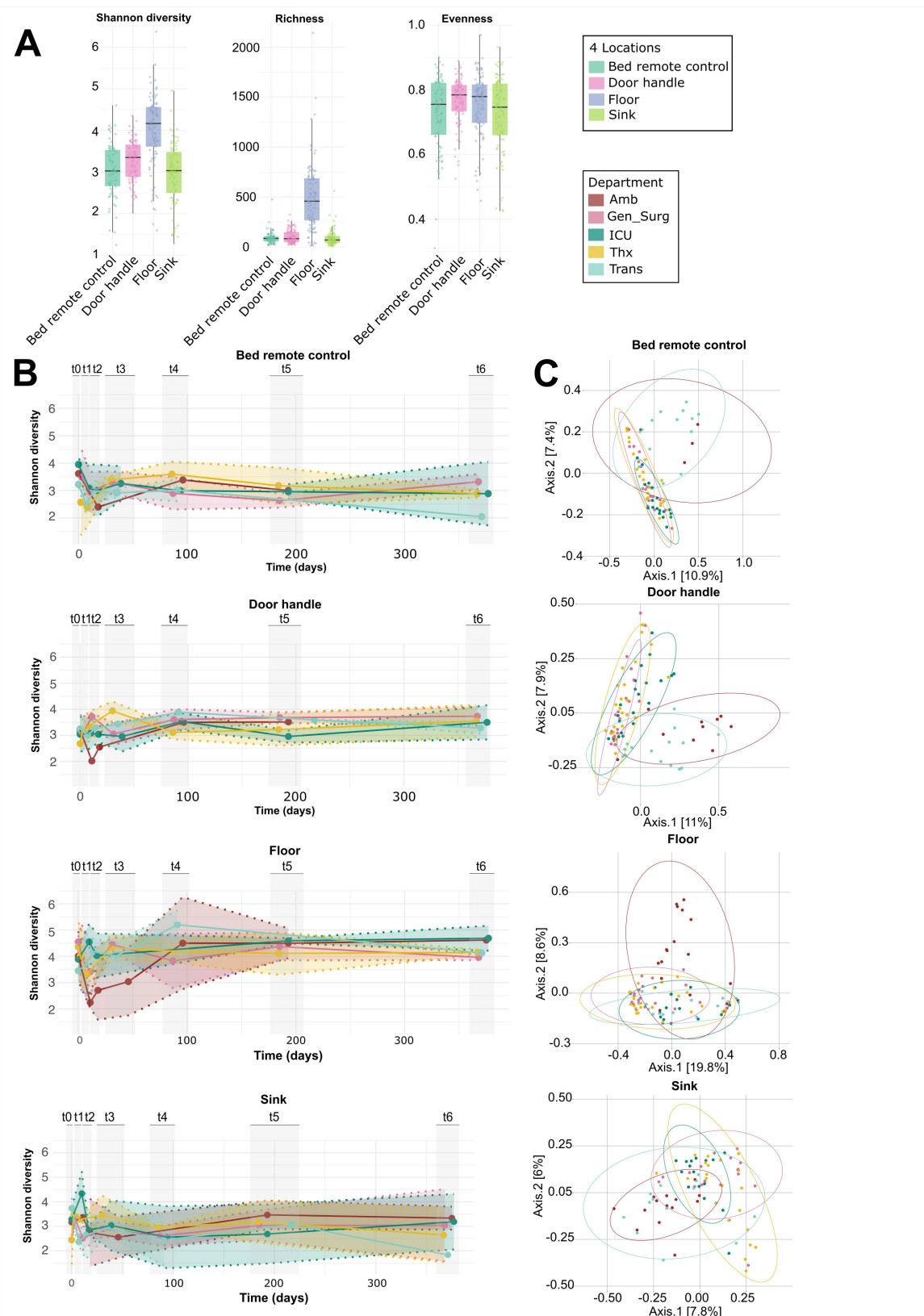

**FIG 4** Diversity across sampling locations in all departments. (A) Alpha diversity of four locations: Shannon diversity, richness (observed ASVs), and evenness. (B) Shannon diversity over time for bed remote control, door handle, floor, and sink per department. (C) PCoA showing beta diversity of the same four locations across all departments.

several surfaces before patient occupancy. Both genera are well known for their ability to withstand desiccation, nutrient limitation, and cleaning procedures and include important MDR species such as *Acinetobacter baumannii* and *Pseudomonas aeruginosa* (80–85). Their early prevalence reflects their ecological versatility, whereas their later decline suggests competitive replacement by human-associated taxa introduced after hospital operation began (86, 87). In contrast, *Corynebacterium* and *Staphylococcus*, commensals of human skin that can also act as opportunistic pathogens (88–94), increased after t1, indicating that direct patient and staff contact is a key driver of surface colonization.

At t0, before the hospital became operational, *Paracoccus* was found to be highly abundant in Gen_Surg and ICU. This could indicate that *Paracoccus* is rather a signature taxon for built environments without clinical context and does not withstand common clinical conditions, such as rigorous cleaning, limited microbial input, low amount of nutrients, and dry conditions. In contrast to *Paracoccus*, *Finegoldia* appeared just in the later time points, meaning that its origin may lie in the input of clinical routine and might have advantages in surviving the harsh cleaning regimes. *Finegoldia* is not a core member of the hospital microbiome but rather seen as a commensal on human skin and mucosal surfaces (95–98). Yet, *F. magna* is an opportunistic pathogen that can cause hospital-acquired infections in immunocompromised people or in the context of surgical procedures (95, 96).

The distinct beta diversity pattern of Amb and Trans (Fig. 3B) could be explained by the high relative abundances of *Pseudomonas* in these two departments at all time points (Fig. 5A). Over time, *Acinetobacter* decreased across all departments except for Trans, where it increased at t5 and decreased again. *Pseudomonas* was particularly abundant at early time points (e.g., t0: Amb, Trans; t1: Trans; t2: Amb), but showed a general decreasing trend over time (Fig. 5A). Similar patterns for *Acinetobacter* and *Pseudomonas* were reported by Lax et al. (33). However, an exception was observed at t1 in Amb, where *Pseudomonas* accounted for approximately 40% of the community (Fig. 5A). Notably, *Staphylococcus* began to increase in relative abundance starting at t2 (approximately 1 week after the hospital became operational) (Fig. 5A).

In the newly built hospital departments, microbial communities developed dynamically across different surface types following hospital occupancy. Before the hospital got operational, *Acinetobacter* and *Pseudomonas* had the highest relative abundance in samples from the bed remote control (Fig. 5C). Similar to beta diversity patterns for door handle samples, Gen_Surg, ICU, Thx, and Trans exhibited relatively stable microbiome compositions over time. In contrast, the Amb department showed high relative abundances of *Pseudomonas* during the first three time points (t0, t1, and t2) and a particularly high abundance of *Acinetobacter* at t1. Across all departments, *Acinetobacter* displayed a consistent decline in relative abundance over time on door handles (Fig. 5C). At early time points (t0–t2), *Acinetobacter* dominated floor samples but decreased over time. Beginning at t2, *Corynebacterium* increased in relative abundance, followed by a rise in *Staphylococcus* after t1, coinciding with the occupation of rooms by patients (Fig. 5C). A distinct microbial signature was observed in sink samples, likely influenced by their wet environment, distinguishing them from other surface types. At t4, sinks in ICU, Thx, and Trans exhibited a high relative abundance of *Streptococcus*. The enrichment of *Streptococcus* in sink-associated communities is consistent with its origin in the human oral and respiratory tract, suggesting droplet-mediated dispersal or washing-related transfer into moist niches (99–101). Its appearance in later time points may therefore reflect increased patient activity and routine clinical care, including oral hygiene practices (8, 102). *Paracoccus* was prevalent in Amb and Trans at t0–t3 but subsequently declined (Fig. 5C).

Overall, these findings suggest a dynamic shift in microbial communities following hospital occupancy, marked by the gradual replacement of early environmental colonizers such as *Acinetobacter* and *Pseudomonas* with human-associated bacteria like *Staphylococcus* and *Corynebacterium* (1, 30, 33, 103). While *Acinetobacter* and

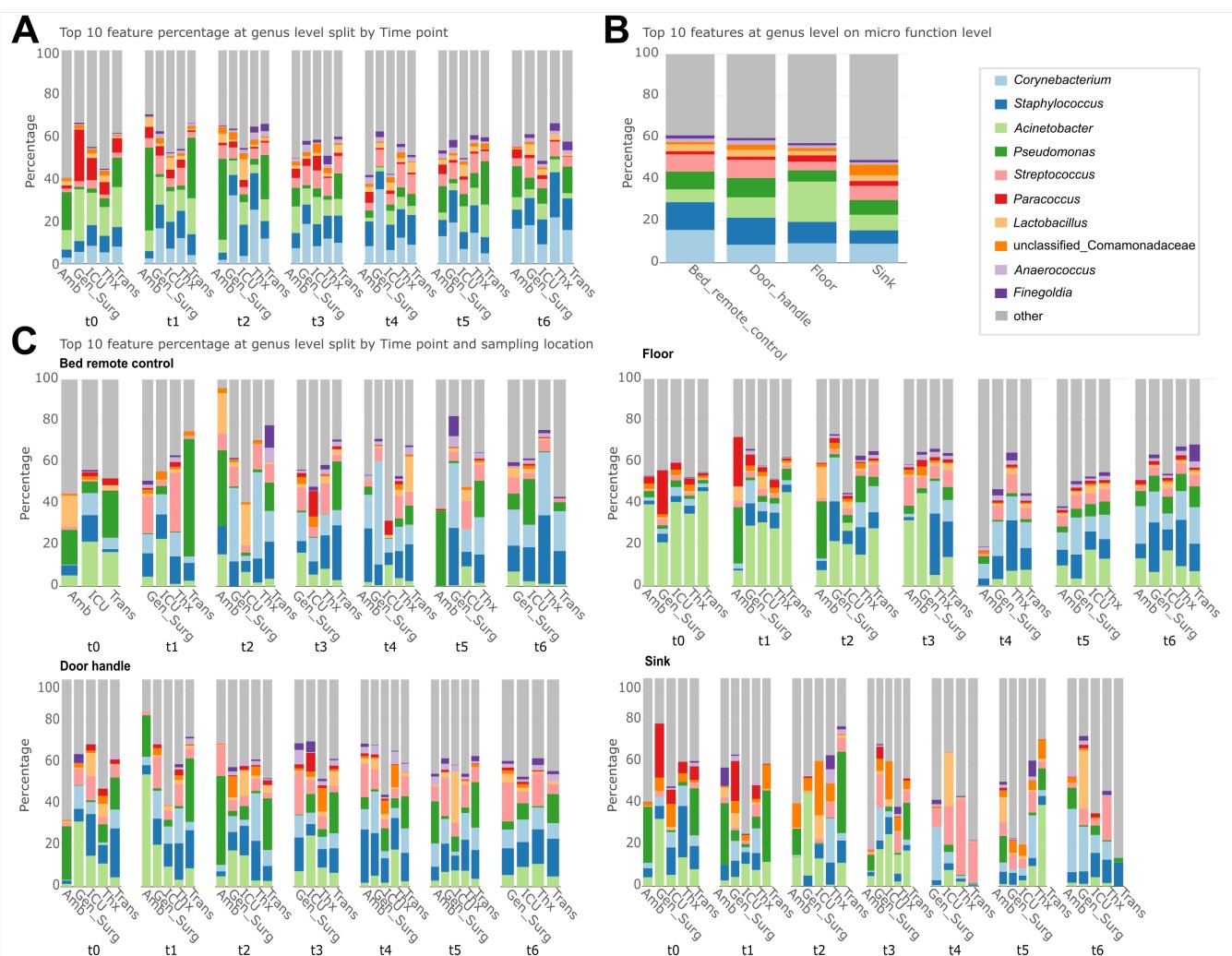

**FIG 5** Relative abundance of the top 10 bacterial genera. (A) Over time per department. (B) Across the four locations sampled in all departments (bed remote control, door handle, floor, and sink). (C) Over time for the same four locations across all departments.

*Pseudomonas* initially dominated multiple surfaces, their abundance generally declined over time, with site-specific and department-specific exceptions influenced by human activity and environmental conditions. In contrast, *Staphylococcus* and *Corynebacterium* became increasingly prevalent, particularly on frequently touched surfaces, indicating potential colonization linked to human presence (1, 104–107). Each surface type showed unique microbial trends shaped by factors such as frequency of human contact (e.g., bed remote controls and door handles), environmental exposure (e.g., floors), and moisture availability (e.g., sinks). The distinct composition of sink-associated microbiota (108, 109) and the persistence of opportunistic pathogens like *Staphylococcus* (110–112) highlight the complex interplay between microbial persistence, human occupancy, and surface characteristics and underscore the need for targeted hygiene protocols to mitigate infection risks.

## Key taxa identified by differential abundance analysis and supervised machine learning show distinct patterns across departments and sampling locations

While performing differential abundance analysis (69), we could identify three key taxa, namely *Staphylococcus*, *Pseudomonas*, and *Acinetobacter* (Fig. 6A, MaAsLin2

*Acinetobacter*: t2, t4, t5 $q < 0.05$; *Staphylococcus*: t3 and t6 $q < 0.05$; and *Pseudomonas*: t4 and t6 $q < 0.05$; Table S4). These taxa were not only among the most abundant in the data set (both macro- [department] and micro- [location] functional levels), but they also stood out as key features in our machine-learning-based classification models (Fig. S4 and S5; Table S5). Along with *Corynebacterium* and *Streptococcus*, they were the most frequent and best predictors of both micro-functional and macro-functional levels, underscoring their ecological and diagnostic relevance across the hospital environment (Fig. S4 and S5).

The abundance of these taxa differed over time: *Staphylococcus* showed a clear trend toward longitudinal homogenization over time. *Pseudomonas* displayed relatively stable abundances, although initial time points (t0–t2) exhibited more variability, which diminished in later time points (t3–t6), suggesting increasing homogeneity. Additionally, Amb and Trans consistently differed from the other departments in their *Pseudomonas* profiles (Fig. 5A and 6). In contrast, the relative abundance of *Acinetobacter* declined over the course of the year and remained relatively homogeneous across all five departments (Fig. 6B). *Corynebacterium* exhibited department-specific behavior. In Gen_Surg and Thx, its abundance was notably elevated at t3 and again at t5 in Gen_Surg compared to the other departments (Fig. 6B). Similar patterns were seen in the sample classification analysis, where *Corynebacterium* emerged as a strong predictor of department and surface type, reinforcing its importance in shaping department-specific microbial profiles.

At a micro-functional level (locations), *Staphylococcus* abundance decreased over time in sink samples, while remaining elevated in door handle and bed remote control samples (Fig. 6C). *Pseudomonas* showed a similar distribution across bed remote control, door handle, and floor samples, while there was a distinct pattern for sink samples. For the latter, abundance initially declined, then slightly increased over time, though remaining lower than in the other three sampling locations (Fig. 6C). Abundances of *Acinetobacter* decreased consistently over time across all four sampling locations, particularly in floor samples, where higher levels were observed up to t3, followed by homogenization (Fig. 6C). This suggests that cleaning practices may have played a role in reducing the presence of *Acinetobacter* (1). For example, probiotic-based cleaning agents can reduce HAI-causative agents like *A. baumannii* by up to 90% more than conventional disinfectants (5). *Corynebacterium*, a taxon commonly associated with human skin, was particularly abundant in bed remote control samples and showed an increasing trend in floor samples over time. These trends, combined with classification output, reinforce the increasing influence of human contact over time on the microbiome of frequently touched surfaces (1, 14, 30).

Overall, at the micro-functional level, *Acinetobacter*, as an early environmental colonizer, exhibited a decreasing trend in abundance across time points, which has also been reported in other studies (33). Its decline may be partly attributed to the increasing relative abundance of skin-associated microbes introduced by staff and patient traffic. In contrast, skin-associated taxa such as *Staphylococcus* and *Corynebacterium* increased in abundance, particularly on high-contact surfaces like bed remote controls and door handles. Beyond these occupancy-driven effects, microbial competition may also play an important role: *Staphylococcus* and *Corynebacterium* are natural skin commensals that can outcompete environmental bacteria such as *Acinetobacter*, especially when the skin barrier remains intact and the microbiome is less disrupted by antibiotics or invasive procedures (86, 87). These commensals can inhibit pathogen colonization either through direct competition for resources or by modulating local immune responses.

Then we investigated dominant taxa contributing to variation in beta diversity and visualized them in biplots (Fig. 7) to indicate their role as potential drivers of microbial community composition across samples (113). These biplots emphasize the influence of various metadata categories, including time points, departments, and sampling locations. The top 10 features represent six taxa, which broadly follow three distinct ecological trends: (i) *Pseudomonas*, associated with biofilm formation; (ii) *Acinetobacter*, with

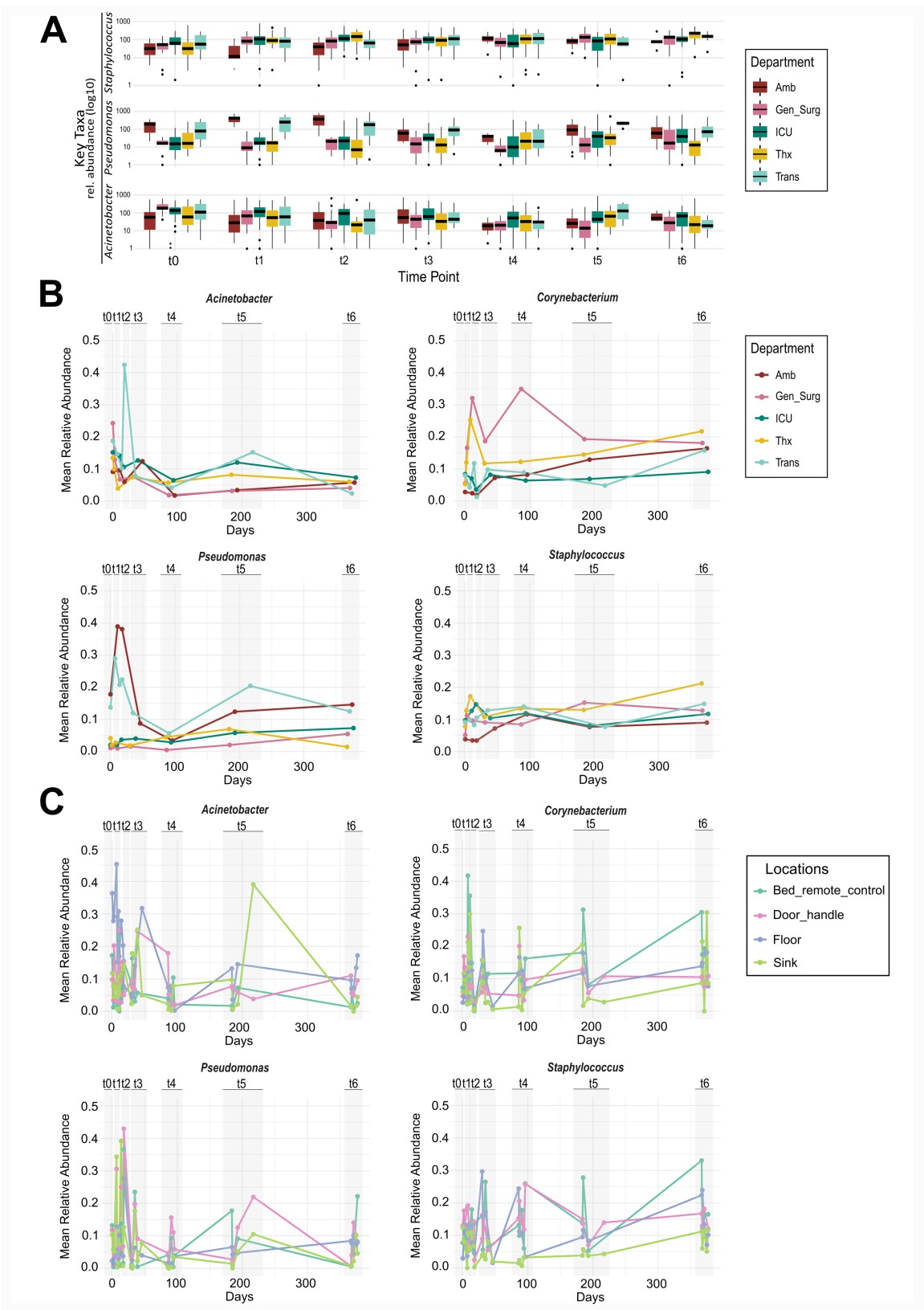

**FIG 6** Key taxa dynamics and volatility. (A) Relative abundance of key taxa *Staphylococcus*, *Pseudomonas*, and *Acinetobacter* over time per department. (B) Volatility plots of *Acinetobacter*, *Corynebacterium*, *Staphylococcus*, and *Pseudomonas* per department over time (metrics in Table S6). (C) Volatility plots of *Acinetobacter*, *Corynebacterium*, and *Pseudomonas* for the four locations sampled in all departments (metrics in Table S6).

strong association to abiotic surfaces; and (iii) human skin-associated bacterial genera, including *Staphylococcus*, *Corynebacterium*, *Streptococcus*, and *Finegoldia* (Fig. 7).

Temporal trends are evident in Fig. 7A: prior to hospital operation (t0), samples cluster closer to the environmental key taxa such as *Acinetobacter* and *Pseudomonas*, whereas after 1 year (t6), they shift toward the skin-associated taxa.

On the macro-functional level (the department level), Amb is predominantly characterized by *Pseudomonas* (Fig. 7B), while ICU samples appear more centrally located, reflecting a mixture of all microbial signatures (Fig. 7C). At the micro-functional level, we observed consistent patterns: floor samples were represented by both *Acinetobacter* and skin-associated taxa; sink samples were more driven by *Pseudomonas*; and bed remote controls were primarily associated with skin-related taxa (Fig. 7D through F).

Together, these findings highlight a dynamic ecological shift in the hospital microbiome over time. Environmental taxa such as *Acinetobacter* and *Pseudomonas* dominated early stages, particularly before the hospital became operational. As human activity increased, skin-associated taxa like *Staphylococcus*, *Corynebacterium*, and *Streptococcus* became more prevalent, particularly on frequently touched surfaces. This indicates a gradual microbial imprinting by hospital staff, patients, and visitors (1, 8, 14, 30, 114). Similar transitions have been reported in other hospital microbiome studies, where early environmental colonizers gave way to human-derived taxa as occupancy increased (34, 42, 115). The convergence of results from both differential abundance and machine-learning-based classification strongly supports this trajectory of human-associated microbial enrichment and the role of specific taxa in defining spatial and temporal microbial dynamics within the hospital.

To assess microbial viability, we collected additional samples from eight of the same sampling locations and treated them with propidium monoazide (PMA) prior to DNA extraction and sequencing (Fig. 1; Fig. S1). PMA is a chemical that selectively penetrates dead or membrane-compromised cells and binds to free DNA, preventing its amplification during sequencing. As such, PMA-treated samples are enriched for DNA from intact (likely viable) cells, whereas untreated samples (non-PMA) reflect the total community, including both viable and dead cells.

Shannon diversity was consistently lower in PMA-treated samples compared to non-PMA samples (Fig. S6A), suggesting that a notable fraction of the detected community in non-PMA samples originated from non-viable cells. Among the key taxa, *Pseudomonas* exhibited significantly higher relative abundances in PMA-treated samples, particularly in Amb and Trans, indicating its potential to persist as an intact and possibly active member of the surface microbiota in these areas (Fig. S6B). This observation further underscores the unique microbial profiles of Amb and Trans compared to the other departments. One possible explanation for the survival of *Pseudomonas* in PMA-treated samples is its ability to form biofilms, which provide protection against environmental stressors and cleaning procedures. *P. aeruginosa*, a model organism for biofilm research, is known for robust biofilm formation that contributes to its persistence on medical equipment, resistance to antibiotics, and role in chronic infections (116–123).

In contrast, *Corynebacterium* and *Staphylococcus* were consistently more abundant in non-PMA samples across all five departments, suggesting that a considerable portion of their detected DNA may have originated from non-intact or dead cells (Fig. S6B). This may reflect frequent deposition from human skin or the environment, with limited long-term survival on hospital surfaces.

*Acinetobacter* showed a more variable pattern: it was more abundant in PMA-treated samples from Amb and Trans, similar to *Pseudomonas*, but also maintained higher abundances in PMA samples from Gen_Surg, ICU, and Thx (Fig. S6B). This widespread viability suggests that *Acinetobacter* is a robust colonizer of hospital surfaces across departments, which may be relevant given its known resilience and association with HAIs. Similar to *Pseudomonas*, *Acinetobacter* can establish and persist within biofilms, supporting its survival on hospital surfaces (4, 124, 125). Previous studies demonstrated that Acinetobacter can remain viable for prolonged periods in both moist environments

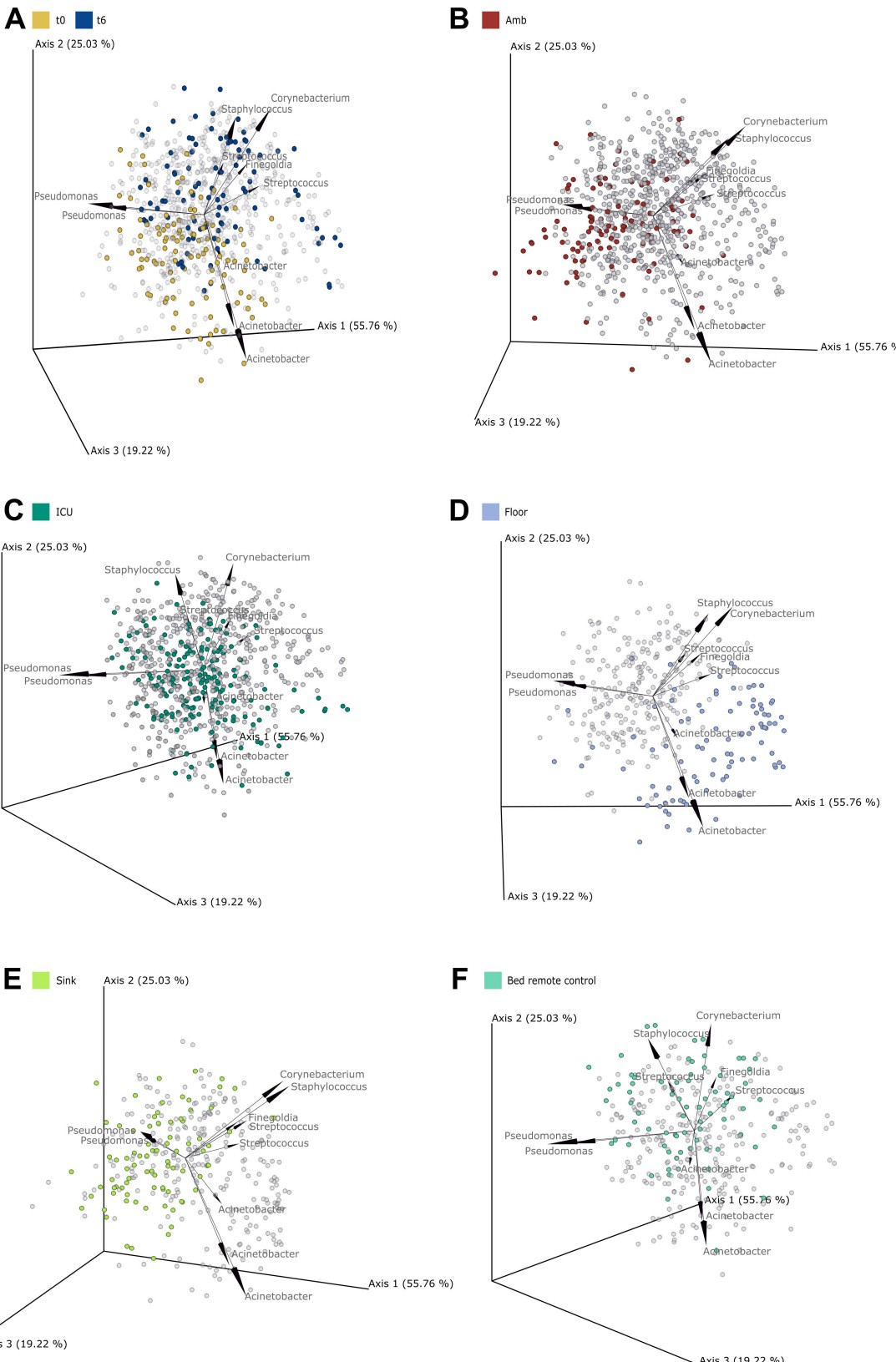

**FIG 7** PCA biplot of top 10 bacterial genera. 3D PCA showing genera with the strongest contributions (arrows). (A) Samples at t0 and t6 (all other time points shown in gray). (B) Amb department, (C) ICU department, (D) floor, (E) sink, and (F) bed remote control samples.

and on dry surfaces such as formica, ceramic, stainless steel, rubber, and polyvinyl chloride (124). Biofilm formation on abiotic surfaces, including medical devices, is a key factor in its persistence, antibiotic resistance, and clinical relevance as a hospital-acquired pathogen (116–120, 123, 126).

To gain further insights, we examined two sampling locations, sink and bed remote control, that were sampled in all departments both with and without PMA treatment (Fig. S6C and D). Interestingly, PMA-treated sink samples showed higher diversity than untreated samples, in contrast to the general trend, indicating a potentially diverse viable community in sink-associated niches. In PMA-treated sink samples, *Acinetobacter* and *Pseudomonas* again dominated, while *Corynebacterium* and *Staphylococcus* were more abundant in non-PMA samples (Fig. S6C and D). A similar pattern was observed for the bed remote control (Fig. S6C and D).

These patterns suggest that *Pseudomonas* and *Acinetobacter* are more likely being represented by intact populations on hospital surfaces, while *Corynebacterium* and *Staphylococcus* may often reflect transient or relic DNA signals from human-associated contamination and/or show a higher sensitivity to the cleaning regimens.

## The way staff and patients interact with surfaces shapes the culturable bacterial fraction of the microbiome in the hospital

In total, 933 contact plates (casein-soy peptone agar; CASO) were employed to monitor a few selected culturable microbes on hospital surfaces across time points (Table S7). A total of 39 samples (4%) were excluded from the comparative analysis due to sampling inconsistencies with the amplicon sequencing data set. Of the remaining 894 samples, 10% showed no microbial growth, while 8% exceeded the threshold of 300 CFUs. Among these high-CFU samples, *Staphylococcus* together with aerobic spore formers were the most frequently identified organisms. Notably, the majority of these high-burden samples originated from the Gen_Surg (38%) and Thx (32%), highlighting these areas as potential hotspots for elevated microbial load.

The most frequently detected microorganisms were *Staphylococcus*, aerobic spore formers, and molds. For further analysis, we focused on samples where *Staphylococcus* was the sole detected organism (representing 34% of the entire data set), as these likely reflect direct human contact with surfaces following hospital operation. In contrast, 39% of samples included *Staphylococcus* alongside one additional microbial group, and 8% included two additional types, pointing to more diverse or potentially older contamination events. Only 0.2% of samples showed *Staphylococcus* with three other taxa.

We observed no consistent increase or decrease in CFUs over time, either at the macro-functional level (across departments) or the micro-functional level (sampling locations), indicating fluctuating contamination levels throughout the year (Fig. 8). This pattern suggests that colonizing events are episodic and likely influenced by variable human activity and cleaning frequency prior to respective sampling activities rather than following a steady longitudinal trend (1, 30). Importantly, the CFU threshold of 300 was not exceeded in any department for *Staphylococcus*-only samples, reinforcing the general success of hygiene protocols. However, floor samples at t6 and toilet flush buttons at t4 did reach the threshold, highlighting this site as a persistent hotspot for microbial accumulation (Fig. 8).

Interestingly, in the Gen_Surg department, no *Staphylococcus*-only samples were observed at t1; all isolates were co-detected with aerobic spore formers (Table S7). This may point to department-specific cleaning practices or environmental differences influencing microbial survival and recovery. For Trans, no cultivation-based samples were taken at t6.

When examining location-specific trends, no clear temporal patterns emerged, either in aggregate CFU counts or in the presence of *Staphylococcus* alone (Table S7). Again, no clear temporal patterns were evident. This reinforces the notion that surface contamination is highly dynamic and influenced by immediate human activity (1, 30), rather than long-term structural differences or time-dependent trends. Furthermore, the

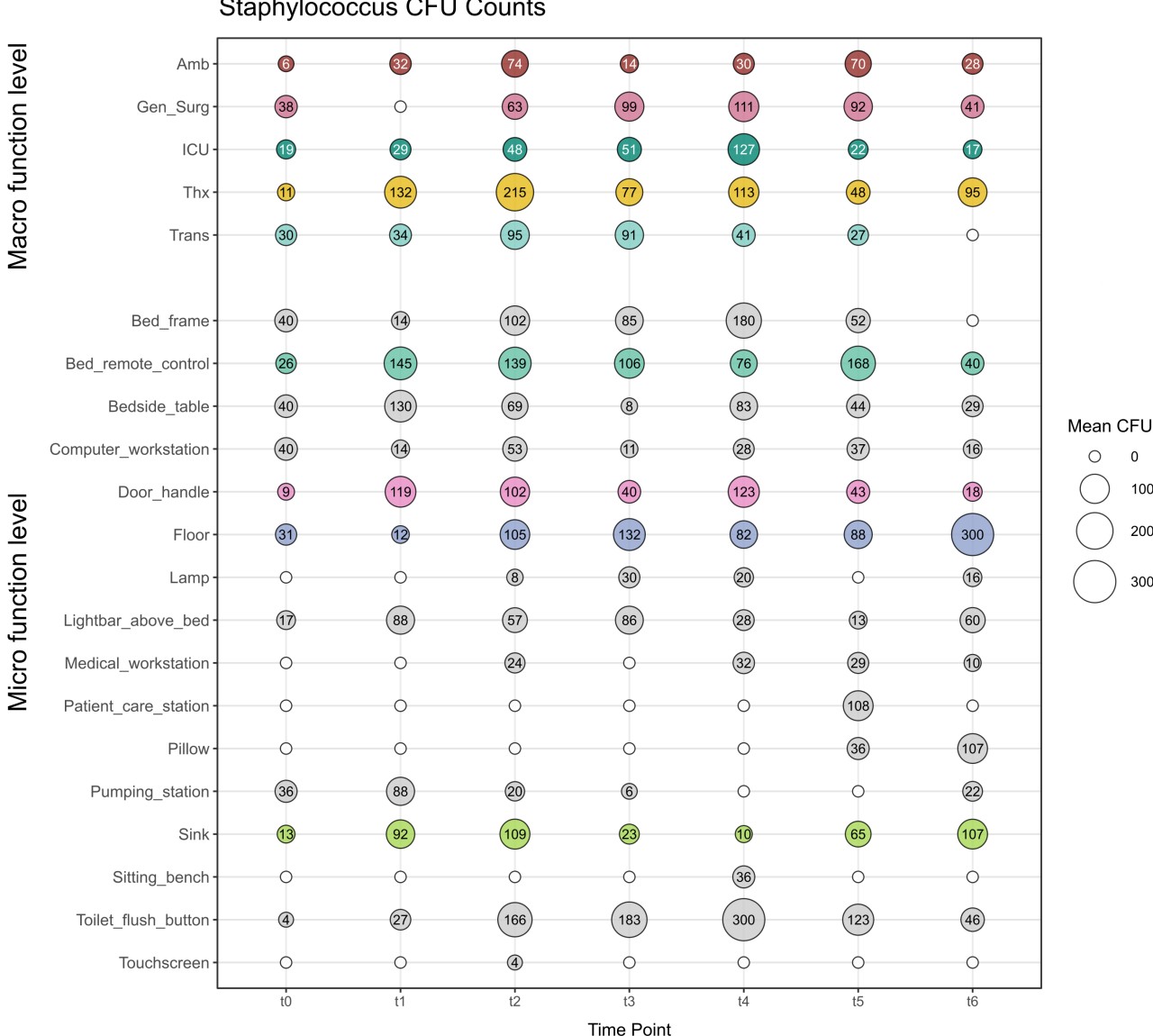

**FIG 8** *Staphylococcus* CFUs across locations. Bubble plot of mean CFU where *Staphylococcus* was exclusively detected, separated by macro-functional and micro-functional levels. White circles indicate no samples; colored circles indicate locations sampled in all departments (bed remote control, door handle, floor, and sink).

inconsistent patterns might also be the result of complex dynamics of contamination (by direct human contact) and removal due to cleaning protocols (decontamination success rates could vary for different surface materials), in dependence on the timely distance to the sampling event. It also supports our 16S rRNA gene-based observations of stable microbiome composition over time, particularly with the dominance of human skin-associated taxa like *Staphylococcus*, *Corynebacterium*, and *Streptococcus* after the hospital became operational.

Together, these cultivation-based results corroborate our sequencing findings, providing evidence that human-associated microbes dominate surface communities in the hospital environment, and that their presence, especially *Staphylococcus*, is a reliable proxy for recent human contact. However, the lack of temporal trends in CFUs under-

scores the complexity of microbial dynamics in built environments, where contamination and removal are in constant interplay.

## Hospital hygiene prevents the spread of potentially predicted pathogens differently on macro-functional and micro-functional levels

For the entire data set, we observed a significant decrease in aerobic (lin. mod. $P$ = 5.1E−05; Spearman's corr. $P$ = 3.51E−6), biofilm-forming (lin. mod. $P$ = 4.7E−9; Spearman's corr. $P$ = 2.81E−13), Gram-negative (lin. mod. $P$ = 8.3E−8; Spearman's corr. $P$ = 2.43E−11), and potential pathogenic (lin. mod. $P$ = 5.32E−6; Spearman's corr. $P$ = 3.81E−8) phenotypes, whereas anaerobic (lin. mod. $P$ = 1.11E−8; Spearman's corr. $P$ = 3.68E−12) and Gram-positive (lin. mod. $P$ = 8.3E−8; Spearman's corr. $P$ = 2.43E−11) phenotypes experienced a significant increase over time (66).

The sampled departments (macro-functional level) differed significantly for the following phenotypes: aerobic (highest in Trans, lowest in Gen_Surg; pairwise Mann-Whitney-Wilcoxon tests FDR corr. $P$ = 5.2E−7; Kruskal-Wallis test all groups $P$ = 1.3E−6), biofilm-formers (highest in Gen_Surg, lowest in ICU; pairwise Mann-Whitney-Wilcoxon tests FDR corr. $P$ = 5.9E−3; Kruskal-Wallis test all groups $P$ = 7.4E−4), Gram-negative (highest in Amb, lowest in Gen_Surg; pairwise Mann-Whitney-Wilcoxon tests FDR corr. $P$ = 1.7E−7; Kruskal-Wallis test all groups $P$ = 3.4E−13), and potential pathogens (highest in Trans, lowest in Gen_Surg; pairwise Mann-Whitney-Wilcoxon tests FDR corr. $P$ = 3.5E−10; Kruskal-Wallis test all groups $P$ = 6.4E−11, Fig. S7).

Over time, most departments showed a significant decrease in aerobic, biofilm-forming, Gram-negative, and potentially pathogenic phenotypes. However, samples from Amb and, in particular, the ICU deviated from these trends. Hence, Amb showed stable proportions of biofilm-forming phenotypes over time, and in contrast to all other departments in the ICU, aerobic, Gram-negative, and potentially pathogenic phenotypes did not change and were stable over time. Most contributing ASVs belong to the phyla Proteobacteria and Actinobacteria. Over time, the relative proportion of contributing Proteobacteria was reduced in comparison to the stable proportion of contributing Actinobacteria. Hotspot locations for aerobic, biofilm-formers, and potential pathogens were the sampled floor surfaces, while the toilet flush button showed the highest proportions of anaerobic phenotypes, the sink for Gram-negative, and the pillow for Gram-positive phenotypes. Focusing on those locations that were sampled in each department, the same direction was visible for bed remote control, door handle, floor, and sink samples. However, the floor was the only sampling location with a significant decrease of aerobic (lin. mod. $P$ = 0.02; Spearman's corr. $P$ = 6.7E−5), biofilm-formers (lin. mod. $P$ = 6.3E−4; Spearman's corr. $P$ = 1.2E−9), Gram-negative (lin. mod. $P$ = 2.0E−4; Spearman's corr. $P$ = 4.8E−9), and potential pathogenic (lin. mod. $P$ = 0.01; Spearman's corr. $P$ = 7.4E−5) phenotypes.

In general, the maintenance of the hospital surfaces was successful in reducing unwanted phenotypes of its microbiome for most departments except the ICU. Here, the proportion of potential pathogens remained stable, a finding consistent with previous reports that ICU environments often act as reservoirs for persistent contamination (75, 127–129). This could either indicate that conventional cleaning regimes are hitting a wall of diminishing returns, or a continuous reseeding event from other less frequently monitored environments such as other devices, staff, or patient surfaces (33, 34). It seems as if the cleaning regime in place creates niches that can be colonized by anaerobes and Gram-positive bacteria, as also suggested by earlier studies highlighting the selective pressures of hospital cleaning (130).

## Conclusion

Our longitudinal study of microbial dynamics across five newly opened hospital departments uncovered a clear two-phase transition in surface microbiomes. In the initial phase—prior to patient arrival—surfaces were dominated by environmental taxa

such as *Acinetobacter* and *Pseudomonas*. As human activity increased, we observed a gradual shift toward human-associated genera, such as *Staphylococcus* and *Corynebacterium*, on frequently touched surfaces. Together, these observations reinforce our central hypothesis that the functional role of hospital interfaces might be key to an adaptive hygiene concept. Importantly, our findings highlight that humans act as the main drivers of hospital microbiome development, introducing and shaping microbial communities through patient occupancy, staff movement, and contact with surfaces (1, 8, 14, 30, 114).

At the macro-functional (department) level, distinct microbial profiles emerged despite comparable overall alpha diversity. In most departments, aerobic, Gram-negative, and potentially pathogenic phenotypes declined over time. However, the ICU stood out: its access restrictions coincided with stable levels of these microbes, suggesting that highly confined environments pose particular challenges for reducing persistent microbial loads. In contrast, the Amb and Trans exhibited greater fluctuations in community composition, clustering apart from other departments (Gen_Surg, Thx, and ICU).

Zooming in further, we found that micro-functional niches (specific sampling locations) had a stronger impact on community composition than their departmental affiliation. This emphasizes the need to adapt microbial monitoring not only by department type but also by the characteristics of individual surfaces and equipment.

While the observed recolonization events in our study are not questionable *per se* (as they could also increase antagonistic potential against pathogens in the hospital environment), they still raise the question of whether alternative locations should not be subject to a more precise, targeted, differentiated clean-up and/or microbial monitoring (21). Hence, department-specific (macro-functional) microbial maintenance and regular observation of its dynamic microbial sources from staff and patients might help to identify early colonization events by hazardous microbes (131, 132). One approach could be to integrate pulsed-UV or vaporized hydrogen peroxide sessions after conventional cleaning only in highly confined areas (e.g., ICU) (131), while employing surface-friendly probiotic sprays with *Bacillus* sp. to outcompete residual biofilm formers in less confined areas (e.g., Amb) (133, 134). Detergents containing *Bacillus* spores have been shown to reduce surface contamination by pathogens, including MDR organisms, in multicenter studies (37, 135), underlying the relevance of microbiome-informed cleaning for antimicrobial resistance (AMR) prevention.

Our study primarily used 16S rRNA gene amplicon sequencing with EMP primers, which are designed for broad environmental sampling. While this approach may not fully capture skin-specific taxa, limit species-level resolution, and exclude important microbiome components such as fungi and viruses, it effectively revealed overall bacterial community dynamics and trends in potentially pathogenic taxa. Future studies employing shotgun metagenomic sequencing could provide species-level and strain-level resolution, functional insights into AMR (30, 136) and virulence determinants, and a more comprehensive view of hospital microbiomes across all microbial kingdoms. Moreover, including air, water, or patient/staff microbiomes could provide a more comprehensive understanding of hospital microbial ecology. Despite these limitations, our work highlights the importance of longitudinal monitoring of the hospital microbiome. Furthermore, integrating microbial phenotype surveillance with quantitative approaches (e.g., qPCR for resistance genes and virulence markers) may help to define high-risk micro-functional locations for targeted interventions, such as antimicrobial-coated flooring in high-traffic zones, filters or self-disinfecting materials in sinks and toilets, or replaceable daily changed covers for pillows and soft furnishings (137).

In the long term, qualitative and quantitative temporal and spatial measurements could be used to establish predictive risk maps trained by phenotype loads versus cleaning frequency in machine-learning models (138). Such approaches could support the development of adaptive hygiene strategies that safeguard patients while minimizing the risk of antimicrobial resistance spread in healthcare environments.

## ACKNOWLEDGMENTS

We appreciate the input and support of Georg Steindl, Martin Josef Koefer, Frederike Reischies, and Gebhard Feierl. We also thank the Core Facility Molecular Biology (ZMF, Medical University Graz, Austria) for library preparation and sequencing.

We thank Stadt Graz for funding.

## AUTHOR AFFILIATIONS

[1]Diagnostic and Research Institute of Hygiene, Microbiology and Environmental Medicine, Medical University of Graz, Graz, Austria
[2]Institute for Hospital Hygiene and Microbiology, Graz, Austria
[3]Division of Immunology, Medical University of Graz, Graz, Austria
[4]Research Unit for Safety and Sustainability in Health Care, Division of Plastic, Aesthetic and Reconstructive Surgery, Medical University of Graz, Graz, Austria
[5]BioTechMed Graz, Graz, Austria

## AUTHOR ORCIDs

Viktoria Weinberger  http://orcid.org/0009-0001-7787-0755
Charlotte Neumann  http://orcid.org/0000-0003-0034-4199
Christina Kumpitsch  http://orcid.org/0000-0002-2077-2839
Stefanie Duller  http://orcid.org/0000-0002-0927-3270
Christine Moissl-Eichinger  http://orcid.org/0000-0001-6755-6263
Alexander Mahnert  http://orcid.org/0000-0001-7083-8894

## FUNDING

| Funder | Grant(s) | Author(s) |
| --- | --- | --- |
| City of Graz | | Kaisa Koskinen |

## AUTHOR CONTRIBUTIONS

Viktoria Weinberger, Formal analysis, Investigation, Methodology, Writing – original draft | Charlotte Neumann, Supervision, Writing – review and editing | Christina Kumpitsch, Investigation, Visualization, Writing – review and editing | Stefanie Duller, Investigation, Methodology, Resources | Tejus Shinde, Writing – review and editing | Polina Mantaj, Investigation, Methodology | Laura Schmidberger, Investigation, Methodology | Tamara Zurabishvili, Investigation, Methodology | Isolde Halmer, Investigation, Methodology | Marina Cecovini, Investigation, Methodology | Simone Vrbancic, Investigation, Methodology | Kathrin Pepper, Project administration, Resources | Eva Schmon, Project administration, Resources | Julian Wenninger, Project administration, Resources | Lars-Peter Kamolz, Project administration, Resources | Gerald Sendlhofer, Project administration, Resources | Kaisa Koskinen, Conceptualization, Funding acquisition | Christine Moissl-Eichinger, Conceptualization, Methodology, Project administration, Resources, Writing – original draft, Writing – review and editing | Alexander Mahnert, Conceptualization, Formal analysis, Investigation, Methodology, Project administration, Supervision, Validation, Writing – original draft, Writing – review and editing

## DATA AVAILABILITY

The raw reads generated in this study have been deposited in the European Nucleotide Archive Database under the accession code PRJEB92068. ASV- tables, metadata, and used scripts are openly available and shared via GitHub (https://github.com/CME-lab-research/MHM-moving-hospital-microbiome). AI was used for language improvement.

## ADDITIONAL FILES

The following material is available online.

### Supplemental Material

**Supplemental figures (Spectrum02178-25-s0001.docx).** Fig. S1 to S7.
**Supplemental tables (Spectrum02178-25-s0002.xlsx).** Tables S1 to S7.

### Open Peer Review

**PEER REVIEW HISTORY (review-history.pdf).** An accounting of the reviewer comments and feedback.

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
