## [Reviewer comments · Microbiology Spectrum]

Microbiology Spectrum

Colonizing the clinic: Tracking bacterial succession and longitudinal dynamics in five new hospital departments over an entire year

Viktoria Weinberger, Charlotte Neumann, Christina Kumpitsch, Stefanie Duller, Tejus Shinde, Polina Mantaj, Laura Schmidberger, Tamara Zurabishvili, Isolde Halmer, Marina Cecovini, Simone Vrabancic, Kathrin Pepper, Eva Schmon, Julian Wenninger, Lars-Peter Kamolz, Gerald Sendlhofer, Kaisa Koskinen, Christine Moissl-Eichinger, and Alexander Mahnert

Corresponding Author(s): Alexander Mahnert, Medizinische Universitat Graz

Review Timeline:

Submission Date:	July 16, 2025
Editorial Decision:	September 2, 2025
Revision Received:	September 28, 2025
Accepted:	October 17, 2025

Editor: Justine Debelius

Reviewer(s): Disclosure of reviewer identity is with reference to reviewer comments included in decision letter(s). The following individuals involved in review of your submission have agreed to reveal their identity: Elisabetta Caselli (Reviewer #2)

Transaction Report:

DOI: <https://doi.org/10.1128/spectrum.02178-25>

Re: Spectrum02178-25 (**Colonizing the clinic: Tracking microbiome succession and longitudinal dynamics in five new hospital departments over an entire year**)

Dear Dr. Alexander Mahnert:

Thank you for the privilege of reviewing your work. Below you will find my comments, instructions from the Spectrum editorial office, and the reviewer comments.

Both reviewers thought the manuscript was relevant. However, they both raised some issues with the manuscript, primarily in the writing. Please provide a stronger foundation and better engage with the existing literature around the build environment. Please also consider arranging the manuscript to integrate the methodological sections around themes or use the headers to clearly distinguish between methodological approaches. As Microbiology Spectrum is a methods last journal, some redundancy between methods and results may be necessary. Finally, please make sure all figures are readable, with a final font size of 8pt or higher in both the main text and supplement.

Revision Guidelines

Sincerely,
Justine Debelius
Editor
Microbiology Spectrum

Reviewer #1 (Comments for the Author):

I appreciate the level of effort to conduct this large longitudinal study of the hospital microbiome. I really enjoyed the use of culturing and PMA to augment traditional sequencing. I would suggest a few areas to improve the manuscript prior to publication:

1. A lot of work has been done in microbiology of the built environment, including in hospitals. It would be great for authors to highlight the knowledge gaps they think they are filling in the abstract and introduction.
2. If possible, Supplemental Figure 1 should be included in manuscript. Understanding the complex sampling plan and location is vital to interpreting the results.
3. Why were certainly locations and facilities sampled? It would help the results and discussion to detail a bit more.
4. There are some limitations with EMP primers for skin microbes. Recommend talking through that, at least as limitation.
5. Results and discussion section lacked context to other work in this area. Introduction was full of references but only four references are included in results/discussion (two back to back sentences).
6. I struggled to follow how results/discussion was written. It seemed to be based on main topic areas, but then was more based on methods in areas. For example, "the way staff and patients interact with surfaces...." seems like it could have been described by sequencing (especially live with PMA) but this section is all culturing). Also wonder about pathogen in next section and analyzing using PMA for intact cells.
7. Consider adding bacteria to title and include a bit more since you didnt look at fungi in this study.

Reviewer #2 (Comments for the Author):

The study examines the temporal dynamics of the hospital surface microbiome across departments by evaluating microbial diversity, potential pathogen sources, and the impact of environmental factors using both uncultured (NGS) and culture-based approaches. The manuscript is highly relevant and could significantly contribute to the field by providing findings that support improved hospital design, cleaning protocols, and infection prevention, ultimately enhancing patient safety. The methods are well described: the authors have appropriately deposited the raw reads in the European Nucleotide Archive Database (accession code PRJEB92068) and provided ASV tables, metadata, and scripts on GitHub, ensuring transparency and reproducibility. The obtained data are presented in a logical and structured manner, allowing for clear interpretation and understanding. However, some minor points require attention.

Specific comments:

Line 23: consider replacing "commenced" with "started"

Line 64: correct the acronym of HAI, with HAIs

Lines 70 and 79: check the space between words and citation in brackets

Lines 101, 102, 103: this paragraph is too brief and largely repeats information already presented in lines 92-99. Consider removing or rephrasing it to avoid redundancy.

Lines 123-128: some information appears redundant, as it refers to the same points mentioned earlier (64-66). It is recommended to rephrase this section to avoid repetition and improve the readability of the text.

Line 132: remove "also"

Lines 140-145: this paragraph is too brief. Consider rephrasing and combining it with the previous one to create a more cohesive and comprehensive section and avoid redundancy.

Line 172: please correct the verb tenses.

Line 176: define the acronym PMA the first time it is mentioned in the "Introduction".

Lines 179-181: the information related to the departments should be included in lines 165-166, as they are first mentioned at that point.

Line 212: please write square meters as m².

Line 235 and 684: specify which types of media were used.

Line 244: correct the typo in "changes : Rather"

Line 257: correct the sentence "cycling conditions were used according to..."

Line 349: correct the typo "identified: After"

Lines 410-412: this paragraph is too brief. Consider to connect it with the previous one

Line 424: consider to substitute the term "locations" with "points"

Figure 4C: increase the resolution and the size of the axis labels to improve readability

Lines 591-596, 618-626, 646-651: add a comparison of your findings with other studies in the literature, where applicable.

Line 812: other studies evaluating the use of Bacillus in innovative sanitation systems could be included (e.g.10.2147/IDR.S194670; 10.3390/ijms24076535)

Conclusion: the paragraph could be further developed, particularly with reference to antimicrobial resistance. It would also be helpful to indicate the study's limitations.

Response to Reviewers

Reviewers Comments and Author Responses:

We appreciate the time of all our reviewers and are thankful for their helpful comments and valid corrections.

Please find our responses below.

Reviewer #1 (Comments for the Author):

I appreciate the level of effort to conduct this large longitudinal study of the hospital microbiome. I really enjoyed the use of culturing and PMA to augment traditional sequencing. I would suggest a few areas to improve the manuscript prior to publication:

1. A lot of work has been done in microbiology of the built environment, including in hospitals. It would be great for authors to highlight the knowledge gaps they think they are filling in the abstract and introduction.

We modified and added additional details to our abstract and introduction to better indicate the knowledge gaps we are addressing with our study design:

“The development of hospital-associated microbial communities over time remains poorly characterized, particularly in terms of how microbial populations dynamically respond to changes in building function, the integration of molecular and cultivation-based data, and the early identification of intervention points for flexible, adaptive microbial control strategies.”

“Despite growing awareness of the role built environments play in shaping microbial communities, the temporal dynamics of microbial colonization in newly opened hospital departments remain poorly understood. In particular, there is limited insight into how microbial populations respond to shifts in building function, such as the transition from construction to clinical use. Existing studies often rely solely on molecular techniques, overlooking the complementary value of using propidium monoazide (PMA) to identify the fraction of intact cells in molecular data or include standard cultivation-based approaches that can still reveal viable and clinically relevant taxa. Moreover, current hygiene strategies tend to be static and generalized, lacking the flexibility to adapt to microbial changes driven by human activity, especially on high-touch surfaces. There is a pressing need to identify early microbial transition points that could serve as targets for proactive and adaptive infection control, rather than reactive interventions. Addressing these gaps is essential for developing precision hygiene concepts that are both ecologically informed and operationally feasible.”

2. If possible, Supplemental Figure 1 should be included in manuscript. Understanding the complex sampling plan and location is vital to interpreting the results. *Supplementary figure S1 was added to the main manuscript and is now shown as Fig. 1.*

3. Why were certainly locations and facilities sampled? It would help the results and discussion to detail a bit more. *A brief, better explanation why these sample locations were chosen was added to the manuscript.*

“The covered departments in this study were: Ambulatory Care Unit (Amb, an outpatient ward- polyclinics), Intensive Care Unit (ICU), General and visceral Surgery (Gen_Surg),

Thorax Surgery (Thx), and Transplant Surgery (Trans). The departments differed in patient turnover and room access: The ICU had the highest level of access restriction, and the longest patient stays, Trans was equipped with an airlock system, whereas Amb had minimal restrictions and only short-term patient visits. In contrast, Gen_Surg, and Thx shared a comparable room structure and organization, which facilitates more direct comparisons of their microbial communities.

Seventeen distinct sampling locations across departments were selected to reflect a range of clinical rooms (e.g., isolation, double, and multi-bedrooms, recovery and examination rooms) and functional surfaces. These locations were chosen because they represent sites with different levels of patient contact, hygiene procedures, and environmental exposure, thereby capturing a broad spectrum of microbial inputs within the hospital setting. The sampling included patient-associated areas (e.g., bed frames, remotes, bedside tables), sanitary facilities (e.g., sinks, toilet flush buttons), and frequently touched zones (e.g., door handles, floors; for a summary of all sampling locations, see Fig. 1)."

4. There are some limitations with EMP primers for skin microbes. Recommend talking through that, at least as limitation.

That's right, we now also mention these limitations in the conclusion of our manuscript.

"Our study primarily used 16S rRNA gene amplicon sequencing with EMP primers, which are designed for broad environmental sampling. While this approach may not fully capture skin-specific taxa, limits species-level resolution, and excludes important microbiome components such as fungi and viruses, it effectively revealed overall bacterial community dynamics and trends in potentially pathogenic taxa."

5. Results and discussion section lacked context to other work in this area. Introduction was full of references but only four references are included in results/discussion (two back to back sentences).

More references and comparisons to other studies have been added to the Results and Discussion section:

"It seems as if the alpha diversity of the indoor microbiome undergoes some fluctuations at the beginning followed by a stabilization over time. Similarly, other studies investigating newly opened hospital departments reported significant increases in alpha diversity immediately after patient occupancy, which then stabilized within a few weeks (8, 71)."

"These differences may also reflect the distinct functions and activities in each department (e.g., patient contact, usage of different clinical areas, and therefore more or less microbial input (24), and variation in sampling sites (micro-functional level). Moreover, the higher beta diversity in Amb and Trans could be driven by increased microbial turnover due to shorter patient stays, greater staff movement, or more diverse environmental inputs. In contrast, Gen_Surg, Thx, and ICU may harbor more stable microbial communities due to more uniform patient populations, structured room usage, and consistent cleaning regimens (24). These patterns underscore the role of human activity as a key driver of hospital surface microbiomes and highlight the need to consider both department-specific practices and micro-functional surface characteristics when designing infection prevention strategies (8, 14, 72)."

"Sink samples displayed the highest similarity in beta diversity across all departments, indicating that sinks, regardless of location, may serve as a shared microbial reservoir, possibly influenced by environmental moisture and water-associated taxa (73–76)."

“This suggests that certain departments harbor more comparable microbial communities, potentially due to similar patient demographics, room usage, or environmental factors, while others, such as Amb and Trans, may be shaped by different interactions or cleaning practices (21, 24, 77–79).”

“This observation is in line with other key publications in this field (33). Acinetobacter and Pseudomonas, two environmentally persistent opportunists, dominated several surfaces before patient occupancy. Both genera are well known for their ability to withstand desiccation, nutrient limitation, and cleaning procedures, and include important multidrug-resistant species such as A. baumannii and P. aeruginosa (80–85). Their early prevalence reflects their ecological versatility, whereas their later decline suggests competitive replacement by human-associated taxa introduced after hospital operation began (86, 87). In contrast, Corynebacterium and Staphylococcus, commensals of human skin that can also act as opportunistic pathogens (88–94) increased after t1, indicating that direct patient and staff contact is a key driver of surface colonization.”

“The enrichment of Streptococcus in sink-associated communities is consistent with its origin in the human oral and respiratory tract, suggesting droplet-mediated dispersal or washing-related transfer into moist niches (99–101). Its appearance in later time points may therefore reflect increased patient activity and routine clinical care, including oral hygiene practices (8, 102).”

“Overall, these findings suggest a dynamic shift in microbial communities following hospital occupancy, marked by the gradual replacement of early environmental colonizers such as Acinetobacter and Pseudomonas with human-associated bacteria like Staphylococcus and Corynebacterium (1, 30, 33, 103). While Acinetobacter and Pseudomonas initially dominated multiple surfaces, their abundance generally declined over time, with site- and department-specific exceptions influenced by human activity and environmental conditions. In contrast, Staphylococcus and Corynebacterium became increasingly prevalent, particularly on frequently touched surfaces, indicating potential colonization linked to human presence (1, 104–107). Each surface type showed unique microbial trends shaped by factors such as frequency of human contact (e.g., bed remote controls, door handles), environmental exposure (e.g., floors), and moisture availability (e.g., sinks). The distinct composition of sink-associated microbiota (108, 109) and the persistence of opportunistic pathogens like Staphylococcus (110–112) highlight the complex interplay between microbial persistence, human occupancy, and surface characteristics and underscore the need for targeted hygiene protocols to mitigate infection risks.”

“This suggests that cleaning practices may have played a role in reducing the presence of Acinetobacter (1). For example, probiotic-based cleaning agents can reduce HAI-causative agents like A. baumannii by up to 90% more than conventional disinfectants (5). Corynebacterium, a taxon commonly associated with human skin, was particularly abundant in bed remote control samples and showed an increasing trend in floor samples over time. These trends, combined with classification output, reinforce the increasing influence of human contact over time on the microbiome of frequently-touched surfaces (1, 14, 30).

Overall, at the micro-functional level, Acinetobacter, as an early environmental colonizer, exhibited a decreasing trend in abundance across time points, which has also been reported

in other studies (33). Its decline may be partly attributed to the increasing relative abundance of skin-associated microbes introduced by staff and patient traffic. In contrast, skin-associated taxa such as Staphylococcus and Corynebacterium increased in abundance, particularly on high-contact surfaces like bed remote controls and door handles. Beyond these occupancy-driven effects, microbial competition may also play an important role: Staphylococcus and Corynebacterium are natural skin commensals that can outcompete environmental bacteria such as Acinetobacter, especially when the skin barrier remains intact and the microbiome is less disrupted by antibiotics or invasive procedures (86, 87)."

"Similar transitions have been reported in other hospital microbiome studies, where early environmental colonizers gave way to human-derived taxa as occupancy increased (34, 42, 115)."

"P. aeruginosa, a model organism for biofilm research, is known for robust biofilm formation that contributes to its persistence on medical equipment, resistance to antibiotics, and role in chronic infections (116–123)."

"Similar to Pseudomonas, Acinetobacter can establish and persist within biofilms, supporting its survival on hospital surfaces (4, 124, 125). Previous studies demonstrated that Acinetobacter can remain viable for prolonged periods in both moist environments and on dry surfaces such as Formica, ceramic, stainless steel, rubber, and polyvinyl chloride (124). Biofilm formation on abiotic surfaces, including medical devices, is a key factor in its persistence, antibiotic resistance, and clinical relevance as a hospital-acquired pathogen (116–120, 123, 126)."

"This pattern suggests that colonizing events are episodic and likely influenced by variable human activity and cleaning frequency prior to respective sampling activities rather than following a steady longitudinal trend (1, 30)."

"This reinforces the notion that surface contamination is highly dynamic and influenced by immediate human activity (1, 30), rather than long-term structural differences or time-dependent trends."

"In general, the maintenance of the hospital surfaces was successful in reducing unwanted phenotypes of its microbiome for most departments except the ICU. Here, the proportion of potential pathogens remained stable, a finding consistent with previous reports that ICU environments often act as reservoirs for persistent contamination (75, 127–129). This could either indicate that conventional cleaning regimes are hitting a wall of diminishing returns, or a continuous reseeding event from other less frequently monitored environments such as other devices, staff or patient surfaces (33, 34). It seems as if the cleaning regime in place creates niches that can be colonized by anaerobes and Gram-positive bacteria, as also suggested by earlier studies highlighting the selective pressures of hospital cleaning (130)."

6. I struggled to follow how results/discussion was written. It seemed to be based on main topic areas, but then was more based on methods in areas. For example, "the way staff and patients interact with surfaces...." seems like it could have been described by sequencing (especially live with PMA) but this section is all culturing). Also wonder about pathogen in next section and analyzing using PMA for intact cells.

Our idea was to first familiarize readers with the study design, then highlight the macro- and micro-functional differences at the alpha and beta diversity levels, and finally find possible explanations for the observed patterns of microbial communities. This required additional methods to sufficiently substantiate our definition of key taxa and then place them in a suitable ecological context. We mentioned specific statistical methods, machine learning, PMA treatment, cultivation-dependent methods, phenotypic predictions, etc. in the text to gradually convince the reader of our interpretations. We hope these explanations make our decision on the structure of our Results/Discussion section now more transparent.

Nevertheless, we have also attempted to simplify the technical language used in some sections (e.g., MaAsLin2, machine-learning, DEICODE, CFU, BugBase) in order to improve readability and better reflect the suggestions made by our reviewers. According to our editor's recommendations we also modified our subheadings to highlight the methodological approaches, and at the same time moved further methodological details to Materials & Methods in order to reduce the disruption to the flow of reading in Results & Discussion caused by individual methodological descriptions.

7. Consider adding bacteria to title and include a bit more since you didnt look at fungi in this study.

We have adjusted our title and now also mention fungi as an obvious limitation of our study in the conclusion of our manuscript.

The title was changed to: "Colonizing the clinic: Tracking bacterial succession and longitudinal dynamics in five new hospital departments over an entire year."

Reviewer #2 (Comments for the Author):

The study examines the temporal dynamics of the hospital surface microbiome across departments by evaluating microbial diversity, potential pathogen sources, and the impact of environmental factors using both uncultured (NGS) and culture-based approaches. The manuscript is highly relevant and could significantly contribute to the field by providing findings that support improved hospital design, cleaning protocols, and infection prevention, ultimately enhancing patient safety. The methods are well described: the authors have appropriately deposited the raw reads in the European Nucleotide Archive Database (accession code PRJEB92068) and provided ASV tables, metadata, and scripts on GitHub, ensuring transparency and reproducibility. The obtained data are presented in a logical and structured manner, allowing for clear interpretation and understanding. However, some minor points require attention.

Specific comments:

Line 23: consider replacing "commenced" with "started"

Corrected.

Line 64: correct the acronym of HAI, with HAIs

Corrected.

Lines 70 and 79: check the space between words and citation in brackets

Corrected.

Lines 101, 102, 103: this paragraph is too brief and largely repeats information already presented in lines 92-99. Consider removing or rephrasing it to avoid redundancy.

The paragraph has been revised and rephrased:

“In newly opened hospitals, surface microbiomes initially resemble those of the outdoor environment, but with the presence of patients and staff, these communities shift towards human-associated taxa, particularly from skin and respiratory tract, including signatures of Corynebacterium, Staphylococcus, Streptococcus, and Acinetobacter (1, 8, 23, 34). Hospital staff can act as vectors, disseminating microbes throughout the facility, while patients imprint their microbial patterns on their immediate surroundings, leading to increasing similarity between room and occupant microbiomes over time (34).”

Lines 123-128: some information appears redundant, as it refers to the same points mentioned earlier (64-66). It is recommended to rephrase this section to avoid repetition and improve the readability of the text.

The paragraph has been revised and rephrased:

“Given their impact on patient safety, the hospital microbiome is of particular concern due to its role as a potential reservoir for opportunistic pathogens. Even in facilities with rigorous infection prevention protocols, environmental contamination can persist and contribute to the spread of HAIs (4, 5, 21, 38–45). Surfaces touched frequently by staff, patients, and visitors can serve as nodes in transmission pathways, enabling pathogens to spread between individuals or persist in the built environment (46). Besides multidrug-resistant (MDR) bacteria such as methicillin-resistant Staphylococcus aureus (MRSA), vancomycin-resistant enterococci (VRE), and carbapenem-resistant Gram-negative bacteria have been detected on hospital surfaces (47–53). Many of these pathogens show alarming levels of antibiotic resistance, which complicates treatment strategies and contributes to poor patient outcomes (21, 38).”

Line 132: remove "also"
Corrected.

Lines 140-145: this paragraph is too brief. Consider rephrasing and combining it with the previous one to create a more cohesive and comprehensive section and avoid redundancy.

The paragraph has been revised and was connected with the previous one:

“Characterizing the microbiome of the hospital environment is therefore crucial for understanding its impact on patient care and infection control. Detailed profiling of these microbial communities not only supports outbreak investigations and improves our knowledge of pathogen reservoirs, but also informs strategies to prevent HAIs and mitigate the spread of resistance (4, 8, 34, 37, 40, 41, 44, 54–56). Importantly, investigating microbial dynamics, colonization patterns, the resistome, and responses to environmental factors such as cleaning, occupancy, and surface materials provides an opportunity to manage these ecosystems more effectively (21).”

Line 172: please correct the verb tenses.
Corrected.

Line 176: define the acronym PMA the first time it is mentioned in the "Introduction".
Corrected.

Lines 179-181: the information related to the departments should be included in lines 165-166, as they are first mentioned at that point.

The paragraph was changed accordingly.

“The study was performed at the University Hospital of Graz, Austria in a newly built surgical department including the following five departments: Ambulatory Care Unit (Amb, an outpatient ward- polyclinics), general and visceral Surgery (Gen Surg), Intensive Care Unit (ICU), Thorax (Thx) and transplant (Trans) surgery. The five departments differ in their confinement levels in means of accessibility for patients and visitors. The ICU department was the most restricted area, whereas Gen Surg, Thx, and Trans were less restricted, and Amb had the easiest accessibility.”

Line 212: please write square meters as m².
Corrected.

Line 235 and 684: specify which types of media were used.
Corrected.

Line 244: correct the typo in "changes : Rather"
Corrected.

Line 257: correct the sentence "cycling conditions were used according to..."
Corrected. Reference was added.

“Cycling conditions were used according to Caporaso et al. (59, 60), initial denaturation for 3 min at 94°C, 35 denaturation cycles for 45 sec at 94°C, annealing 60 sec 50°C, extension 90 sec 72°C, and final extension for 10 min 72°C.”

Line 349: correct the typo "identified: After"
Corrected.

Lines 410-412: this paragraph is too brief. Consider to connect it with the previous one
The paragraph has been revised and connected to the previous one.

“While alpha diversity did not differ significantly on macro-functional level (Fig. 2A), clear differences in beta diversity were observed (Fig. 3B). Amb and Trans samples were more similar to each other with a higher variation, while the other three departments (Gen_Surg, Thx, and ICU) formed tighter clusters (Fig. 3B). The greater variability in Amb may be attributed to both temporal fluctuations and poorer sequencing quality. Furthermore, Amb and Trans had a high sample drop-out during SRS normalization (Amb: 101 samples, Trans: 57 samples, Supplemental Fig. S1 and Supplemental Table 1). These differences may also reflect the distinct functions and activities in each department (e.g., patient contact, usage of different clinical areas, and therefore more or less microbial input (24), and variation in sampling sites (micro-functional level). Moreover, the higher beta diversity in Amb and Trans could be driven by increased microbial turnover due to shorter patient stays, greater staff movement, or more diverse environmental inputs. In contrast, Gen_Surg, Thx, and ICU may harbor more stable microbial communities due to more uniform patient populations, structured room usage, and consistent cleaning regimens (24). These patterns underscore the role of human activity as a key driver of hospital surface microbiomes and highlight the need to consider both department-specific practices and micro-functional surface characteristics when designing infection prevention strategies (8, 14, 72).”

Line 424: consider to substitute the term "locations" with "points"

Thank you for this suggestion. We considered the alternative wording, but decided to retain “locations” to remain consistent with terminology used throughout the manuscript and to avoid confusion with “time points,” which are also an important part of our study.

Figure 4C: increase the resolution and the size of the axis labels to improve readability
Resolution and size of axis labels were improved for the figure, now 5C:

Lines 591-596, 618-626, 646-651: add a comparison of your findings with other studies in the literature, where applicable.

Additional references and comparisons to other studies were added.

“Overall, at the micro-functional level, Acinetobacter, as an early environmental colonizer, exhibited a decreasing trend in abundance across time points, which has also been reported in other studies (33). Its decline may be partly attributed to the increasing relative abundance of skin-associated microbes introduced by staff and patient traffic. In contrast, skin-associated taxa such as Staphylococcus and Corynebacterium increased in abundance, particularly on high-contact surfaces like bed remote controls and door handles. Beyond these occupancy-driven effects, microbial competition may also play an important role: Staphylococcus and Corynebacterium are natural skin commensals that can outcompete environmental bacteria such as Acinetobacter, especially when the skin barrier remains intact and the microbiome is less disrupted by antibiotics or invasive procedures (86, 87). These commensals can inhibit pathogen colonization either through direct competition for resources or by modulating local immune responses.”

“Together, these findings highlight a dynamic ecological shift in the hospital microbiome over time. Environmental taxa such as Acinetobacter and Pseudomonas dominated early stages, particularly before the hospital became operational. As human activity increased, skin-

associated taxa like Staphylococcus, Corynebacterium, and Streptococcus became more prevalent, particularly on frequently-touched surfaces. This indicates a gradual microbial imprinting by hospital staff, patients, and visitors (1, 8, 14, 30, 114). Similar transitions have been reported in other hospital microbiome studies, where early environmental colonizers gave way to human-derived taxa as occupancy increased (34, 42, 115). The convergence of results from both differential abundance and machine learning-based classification strongly supports this trajectory of human-associated microbial enrichment and the role of specific taxa in defining spatial and temporal microbial dynamics within the hospital.”

“Acinetobacter showed a more variable pattern: it was more abundant in PMA-treated samples from Amb and Trans, similar to Pseudomonas, but also maintained higher abundances in PMA samples from Gen_Surg, ICU, and Thx (Supplemental Fig. S6B). This widespread viability suggests that Acinetobacter is a robust colonizer of hospital surfaces across departments, which may be relevant given its known resilience and association with healthcare-associated infections. Similar to Pseudomonas, Acinetobacter can establish and persist within biofilms, supporting its survival on hospital surfaces (4, 124, 125). Previous studies demonstrated that Acinetobacter can remain viable for prolonged periods in both moist environments and on dry surfaces such as Formica, ceramic, stainless steel, rubber, and polyvinyl chloride (124). Biofilm formation on abiotic surfaces, including medical devices, is a key factor in its persistence, antibiotic resistance, and clinical relevance as a hospital-acquired pathogen (116–120, 123, 126).”

Line 812: other studies evaluating the use of Bacillus in innovative sanitation systems could be included (e.g.10.2147/IDR.S194670; 10.3390/ijms24076535)

The use of Bacillus as an addition to the cleaning regime is already mentioned in the conclusion:

“One approach could be to integrate pulsed-UV or vaporized hydrogen peroxide sessions after conventional cleaning only in highly confined areas (e.g., ICU) (131), while employing surface-friendly probiotic sprays with Bacillus sp. to outcompete residual biofilm formers in less confined areas (e.g. Amb)(133, 134). Detergents containing Bacillus spores have been shown to reduce surface contamination by pathogens, including multidrug-resistant organisms (MDROs), in multicenter studies (37, 135), underlying the relevance of microbiome-informed cleaning for antimicrobial resistance (AMR) prevention.”

Conclusion: the paragraph could be further developed, particularly with reference to antimicrobial resistance. It would also be helpful to indicate the study's limitations.

The conclusion has been improved and amended by adding the limitations of the study:

Our study primarily used 16S rRNA gene amplicon sequencing with EMP primers, which are designed for broad environmental sampling. While this approach may not fully capture skin-specific taxa, limits species-level resolution, and excludes important microbiome components such as fungi and viruses, it effectively revealed overall bacterial community dynamics and trends in potentially pathogenic taxa. Future studies employing shotgun metagenomic sequencing could provide species- and strain-level resolution, functional insights into AMR (30, 136) and virulence determinants, and a more comprehensive view of hospital microbiomes across all microbial kingdoms. Moreover, including air, water, or patient/staff

microbiomes could provide a more comprehensive understanding of hospital microbial ecology. Despite these limitations, our work highlights the importance of longitudinal monitoring of the hospital microbiome. Furthermore, integrating microbial phenotype surveillance with quantitative approaches (e.g., qPCR for resistance genes and virulence markers) may help to define high-risk micro-functional locations for targeted interventions, such as antimicrobial-coated flooring in high-traffic zones, filters or self-disinfecting materials in sinks and toilets, or replaceable daily changed covers for pillows and soft furnishings (137).

In the long term, qualitative and quantitative temporal and spatial measurements could be used to establish predictive risk maps trained by phenotype loads vs. cleaning frequency in machine-learning models (138). Such approaches could support the development of adaptive hygiene strategies that safeguard patients while minimizing the risk of antimicrobial resistance spread in healthcare environments.

Re: Spectrum02178-25R1 (**Colonizing the clinic: Tracking bacterial succession and longitudinal dynamics in five new hospital departments over an entire year**)

Dear Dr. Alexander Mahnert:

Your manuscript has been accepted, and I am forwarding it to the ASM production staff for publication. Your paper will first be checked to make sure all elements meet the technical requirements. ASM staff will contact you if anything needs to be revised before copyediting and production can begin. Otherwise, you will be notified when your proofs are ready to be viewed.

Sincerely,
Justine Debelius
Editor
Microbiology Spectrum

Reviewer #1 (Comments for the Author):

My comments have been adequately addressed.

Reviewer #2 (Comments for the Author):

The authors have fulfilled the concerns raised by the reviewer